



# 1 Isotopic constraints on heterogeneous sulfateproduction in Beijing haze

Pengzhen He[1], Becky Alexander[2], Lei Geng[1], Xiyuan Chi[1], Shidong Fan[1], Haicong Zhan[1], Hui Kang[1], Guangjie Zheng[3]†,
Yafang Cheng[3,4], Hang Su[4,3],Cheng Liu[1,5,6],Zhouqing Xie[1,5,6]*
[1]Anhui Province Key Laboratory of Polar Environment and Global Change, School of Earth and Space Sciences, University
of Science and Technology of China, Hefei, Anhui 230026, China.
[2]Department of Atmospheric Sciences, University of Washington, Seattle, WA 98195, USA.
[3]Multiphase Chemistry Department, Max Planck Institute for Chemistry, Mainz 55128, Germany.
[4]Jinan University, Institute for Environment and Climate Research, Guangzhou, Guangdong 511443, China.
[5]Key Lab of Environmental Optics and Technology, Anhui Institute of Optics and Fine Mechanics, Chinese Academy of
Sciences, Hefei, Anhui 230031, China.
[6]Center for Excellence in Urban Atmospheric Environment, Institute of Urban Environment, Chinese Academy of Sciences,
Xiamen, Fujian 361021, China.
*Corresponding to: Zhouqing Xie (zqxie@ustc.edu.cn)
†: Now at: Atmospheric Sciences Division, Brookhaven National Laboratory, Upton, NY 11973, USA.
**Abstract.** Discerning mechanisms of sulfate formation during fine-particle pollution (referred to as haze hereafter)in Beijing
is important for understanding the rapid evolution of haze and for developing cost-effective air pollution mitigation strategies.
Here we present the first observations of the oxygen-17 excess of $PM_{2.5}$ sulfate ($\Delta^{17}O(SO_4^{2-})$) collected in Beijing haze from
October 2014 to January 2015, to constrain possible sulfate formation pathways. Throughout the sampling campaign, the
12h-averaged $PM_{2.5}$ concentrations ranged from 16 to 323 μg m$^{-3}$ with a mean of (141±88 (1σ)) μg m$^{-3}$, with $SO_4^{2-}$
representing 8–25% of $PM_{2.5}$ mass. The observed $\Delta^{17}O(SO_4^{2-})$ varied from 0.1‰ to 1.6‰ with a mean of
(0.9±0.3)‰.$\Delta^{17}O(SO_4^{2-})$increased with $PM_{2.5}$ levelsin October 2014 while the opposite trendswere observed in November
2014 to January 2015. Heterogeneous sulfate production rate ($P_{het}$) on aerosols was estimated to enhance with $PM_{2.5}$ levels,
generally dominating sulfate formation during haze days when cloud liquid water content (LWC) was low. When LWC was
high, however, in-cloud reactions would dominate haze sulfate formationwith a fractional contribution up to 68%. For the
specific mechanisms of heterogeneous oxidation of $SO_2$, chemical reaction kinetics calculations suggest S(IV) (= $SO_2·H_2O$ +
$HSO_3^-$ + $SO_3^{2-}$) oxidation by $H_2O_2$ in aerosol water accounted for 5–13% of $P_{het}$. The relative importance of heterogeneous
sulfate production by other mechanisms was constrained by our observed $\Delta^{17}O(SO_4^{2-})$. Heterogeneous sulfate production via
S(IV) oxidation by $O_3$ was estimated to contribute 21–22% of $P_{het}$ on average. Heterogeneous sulfate production pathways
that result in zero-$\Delta^{17}O(SO_4^{2-})$, such as S(IV) oxidation by $NO_2$in aerosol water and/or by $O_2$ on acidic microdroplets via a
radical chain mechanism, contributed the remain 66–73% of $P_{het}$. The assumption about the thermodynamic state of aerosols



(stable or metastable) was found to significantly influence the calculated aerosol pH (7.6±0.1 or 4.7±1.1, respectively), and
thus influence the relative importance of heterogeneous sulfate production via S(IV) oxidation by $NO_2$ and by $O_2$ on acidic
microdroplets. Our calculation suggests sulfate formation via $NO_2$ oxidation can be the dominant pathway in aerosols at high
pH-conditions calculated assuming stable state while S(IV) oxidation by $O_2$ on acidic microdroplets can be the dominant
pathway providing that highly acidic aerosols (pH ≲ 3) exist . Our results also illustrate the utility of $\Delta^{17}O(SO_4^{2-})$ for
quantifying sulfate formation pathways and its inclusion in models may improve our understanding of rapid sulfate
formation during haze events.
**1 Introduction**

40       Frequent occurrence of haze events in Beijing and throughout the North China Plain (NCP) during cold seasons is a

health threat for round 400 million people living there. High concentrations of $PM_{2.5}$ (particulate matter with an aerodynamic
diameter less than 2.5 μm), of which the daily average can exceed 300 μg m$^{-3}$ during severe haze(He et al., 2014;Jiang et al.,
2015), contribute to cardiovascular morbidity and mortality(Brook et al., 2010;Cheng et al., 2013). As one of the major
components of $PM_{2.5}$, sulfate is of particular concern due to its high concentrations in haze days (Zheng et al., 2015b;Zheng
et al., 2015a) and its key role in the climate system (Seinfeld and Pandis, 2012). Hourly sulfate concentrations can exceed
100 μg m$^{-3}$ and account for up to one quarter of $PM_{2.5}$ mass during severe haze (Zheng et al., 2015a). However, due to the
generally low solar radiation and cloud liquid water content (LWC) during haze (Zheng et al., 2015a;Wang et al., 2014),
conventional sulfate formation via OH oxidation in the gas-phase and from aqueous-phase $SO_2$ (referred to as S(IV) =
$SO_2 \cdot H_2O + HSO_3^- + SO_3^{2-}$) oxidation by $H_2O_2$(McArdle and Hoffmann, 1983), $O_3$(Hoffmann and Calvert, 1985), and $O_2$ via
a radical chain mechanism initiated by transition metal ions (TMIs) in clouds (Ibusuki and Takeuchi, 1987;Alexander et al.,
2009;Harris et al., 2013) cannot explain the observed high sulfate concentrations (Wang et al., 2014). To explain the
observed high sulfate concentrations during haze in Beijing and NCP, recent studies have suggested that heterogeneous
reactions on/in aerosols/aerosol water are potentially important (He et al., 2014;Hung and Hoffmann, 2015;Cheng et al.,
2016;Wang et al., 2016;Zheng et al., 2015a;Zheng et al., 2015b;Wang et al., 2014). In particular, Zheng et al.(2015a) largely
improved the underestimate of modelled sulfate concentrations in 2013 Beijing haze by using a relative humidity (RH)-
dependent uptake coefficient (γ) of $SO_2$ on aerosols, without knowing the specific mechanisms of heterogeneous oxidation of
$SO_2$. Hung and Hoffmann(2015)proposed that rapid S(IV) oxidation by $O_2$ via a radical chain mechanism initiated due to the
speciality of interfacial water on acidic microdroplets (pH ≤ 3) could be responsible for heterogeneous sulfate production in
Beijing haze, while Cheng et al.(2016) suggested that S(IV) oxidation by $NO_2$(Lee and Schwartz, 1982;Clifton et al., 1988)
in aerosol water could be important due to the high RH and $NO_2$ concentrations during severe haze in  NCP. Due to the
strong pH-dependence of these two pathways and the large variability of model calculated aerosol pH in Beijing haze
(Cheng et al., 2016;Wang et al., 2016;Liu et al., 2017), the relative importance of these two pathways is difficult to constrain.





The oxygen-17 excess ($\Delta^{17}O$) of sulfate, defined as $\Delta^{17}O = \delta^{17}O - 0.52 \times \delta^{18}O$ wherein
$\delta^X O = ((^X O/^{16}O)_{sample}/(^X O/^{16}O)_{VSMOW} - 1)$ with X = 17 or 18 and VSMOW referring to Vienna Standard Mean Ocean
Water(Matsuhisa et al., 1978), is a useful tool for estimating the relative importance of different sulfate formation pathways
because each oxidant transfers its $\Delta^{17}O$ signature to the product (Table 1) through $SO_2$ oxidation (Savarino et al., 2000). $SO_2$
has $\Delta^{17}O = 0$‰ due to the rapid isotopic exchange with abundant vapour water whose $\Delta^{17}O$ is near 0‰(Holt et al., 1981).
S(IV) oxidation by $H_2O_2$ and $O_3$ leads to $\Delta^{17}O(SO_4^{2-}) = 0.7$‰ and 6.5‰, respectively, on the basis of $\Delta^{17}O(H_2O_2) =$
1.4‰(Savarino and Thiemens, 1999)and assuming $\Delta^{17}O(O_3) = 26$‰ (Vicars and Savarino, 2014;Ishino et al., 2017). All
other sources of sulfate exhibit $\Delta^{17}O(SO_4^{2-})$ at or near 0‰. Specifically, sulfate directly emitted from natural and
anthropogenic sources or formed by OH and $O_2$oxidation has $\Delta^{17}O(SO_4^{2-})$ values at or near 0‰ (Dubey et al., 1997;Luz and
Barkan, 2005;Lee et al., 2002;Bao et al., 2000). Sulfate produced by $NO_2$ oxidation is suggested to occur either via a radical
chain mechanism (Shen and Rochelle, 1998) or via oxygen-atom transfer from $O_2$(He et al., 2014), resulting in $\Delta^{17}O(SO_4^{2-})$
= 0‰. Once formed, atmospheric sulfate does not undergo further isotopic exchange, thus $\Delta^{17}O(SO_4^{2-})$ will not be altered by
mass-dependent processes such as deposition.
In this work, we report the first observations of $PM_{2.5}$ $\Delta^{17}O(SO_4^{2-})$ during haze events from October 2014 to January
2015 in Beijing, and use them to quantify the relative importance of different sulfate formation pathways.
**2 Materials and Methods**
**2.1 Sampling and atmospheric observations**
A high volume air sampler (model TH-1000C II, Tianhong Instruments Co., Ltd, China) with quartz microfiber filter
(Whatman Inc., UK, pre-combusted at 450°C for 4 h) was used to collect $PM_{2.5}$ samples at a flow rate of 1.05 $m^3$ $min^{-1}$ from
October 2014 to January 2015. The collections lasted for 12 h (08:00–20:00 LT or 20:00–08:00 LT) for each sample. The
sample site is located on the rooftop of the First Teaching Building at the campus of University of the Chinese Academy of
Sciences (40.41°N, 116.68°E, round20 m from the ground) in Beijing, round60 km northeast of downtown. Hourly $PM_{2.5}$,
$SO_2$, $NO_2$ and $O_3$ concentrations were observed at Huairou station (40.33°N, 116.63°E) by Beijing Municipal Environmental
Monitoring Center, which is about 10 km from our aerosol sampling site. The concentration of atmospheric $H_2O_2$ was not
observed in our campaign, but long-term observations from March to November in Beijing shows a good correlation
between $H_2O_2$ concentrations (ppb) and air temperature (T, in °C) according to $[H_2O_2] = 0.1155 \times e^{(0.0846 \times T)}$(Fu, 2014). In the
present study, $H_2O_2$ concentrations is estimated from our measured T with the above empirical equation. Our calculated
$H_2O_2$ concentration based on this formula in October and November 2014 is respectively (0.32±0.08) ppb and (0.17±0.04)
ppb, comparable to the observed values of (0.44±0.18) ppb and (0.38±0.11) ppb, respectively in October and November
2013 (Fu, 2014). Meteorological data including T and relative humidity (RH) were recorded by an automatic weather station
(model MetPak with integrated wind sonic, Gill Instruments Limited, UK). Time reported in this paper is local time (LT =
UTC + 8).



## 2.2 Measurements of ions and isotope ratios

A detailed description of the method for chemical analysis of $NH_4^+$, $K^+$, $Ca^{2+}$, $Na^+$, $Mg^{2+}$, $SO_4^{2-}$, $NO_3^-$ and $Cl^-$ can be found in the literature (Ye et al., 2015). Briefly, ions were extracted from a part (2 cm × 2 cm) of each filter with 20 ml of Millipore water ($\geq$ 18 MΩ) by sonication for 80 min in an ice water bath. Insoluble substances in the extract were filtered with 0.45 μm filters before analysis. Then the pH of filtrates was measured by an ion activity meter (model PXS-215, Shanghai INESA Scientific Instrument Co., Ltd., China). And the ion concentrations were analysed using Dionex ICS-2100 ion chromatograph system (Thermo Fisher Scientific Inc., USA). The preparation and measurements of $\Delta^{17}O(SO_4^{2-})$ were conducted in Isolab at the University of Washington, USA. A detailed description of the method can be found in the literature (Savarino et al., 2001;Geng et al., 2013). Briefly, $PM_{2.5}$ sample filters were dissolved in Millipore water ($\geq$ 18 MΩ) and the insoluble substances were filtered. Pre-packed ion capture cartridges (Alltech Maxi-Clean IC-RP SPE) were used for the first step of removal of organics. Cations in the samples were replaced with sodium using a cation exchange resin and 30% $H_2O_2$ solution was added as the second step of removal of organics. Excess $H_2O_2$ was removed via evaporation and $SO_4^{2-}$ was separated from other ions in solution by ion chromatography. After ion separation, $SO_4^{2-}$ was converted to $Ag_2SO_4$, dried, and then pyrolyzed at 1000°C in an elemental analyzer to form $Ag(s)$, $SO_2(g)$, and $O_2(g)$. The produced gases were carried by He gas to pass through a liquid nitrogen trap to remove $SO_2$, and then a GC to further purify the $O_2$ gas which was finally induced to a mass spectrometer (Thermo Scientific MAT 253). Masses of 32, 33 and 34 of $O_2$ were measured to determine $\delta^{17}O$ and $\delta^{18}O$ and then $\Delta^{17}O$ was calculated. The precision of $\Delta^{17}O$ measurement in this method is ±0.3‰ based on replicate analysis of standards. To quantify the uncertainty in each sample, 30 samples were measured in triplicate, 2 samples in quadruplicate, and 2 samples in duplicate depending on the limitation of sample size. Totally, 10 filters sampled in non-polluted days (NPD, $PM_{2.5} < 75$ μg m$^{-3}$) and 24 filters sampled in polluted days (PD, $PM_{2.5} \geq 75$ μg m$^{-3}$) were analysed.

## 2.3 Estimate of the overall rate of heterogeneous sulfate production

Heterogeneous sulfate production ($P_{het}$) is commonly parameterized in models according to Eq. (1) (Jacob, 2000;Zheng et al., 2015a):

$$P_{het} = \frac{3600 \times 96}{RT} \left( \frac{R_p}{D_g} + \frac{4}{v\gamma} \right)^{-1} S_p [SO_2(g)] \quad (1)$$

where $P_{het}$ is in unit of μg m$^{-3}$ h$^{-1}$, 3600 is a time conversion factor (s h$^{-1}$), 96 is the molar mass of $SO_4^{2-}$ (g mol$^{-1}$), $R$ is the gas constant (0.082 atm L K$^{-1}$ mol$^{-1}$), and $T$ is temperature in K. $R_p$ is the radius of aerosol particles (m), $D_g$ is the gas-phase molecular diffusion coefficient of $SO_2$ (m$^2$ s$^{-1}$), $v$ is the mean molecular speed of $SO_2$ (g) (m s$^{-1}$), $\gamma$ is the uptake coefficient of $SO_2$ on aerosols (unitless), [$SO_2$ (g)] is the gas-phase concentration of $SO_2$ (ppb) and $S_p$ is the aerosol surface area per unit volume of air (m$^2$ m$^{-3}$). The typical tropospheric value of $D_g$ and $v$ is $2 \times 10^{-5}$ m$^2$ s$^{-1}$ and 300 m s$^{-1}$, respectively (Jacob, 2000). Observations of $PM_{2.5}$ mass concentrations (μg m$^{-3}$) and $PM_{2.5}$ mean radius (m) during Beijing haze roughly follows an empirical formula: $R_p = (0.254 \times PM_{2.5} + 10.259) \times 10^{-9}$ (Guo et al., 2014). By using the volume and surface area formulas of a sphere and the mean density of particles ($\rho = 1.5 \times 10^6$ g m$^{-3}$(Guo et al., 2014)), $S_p$ can be estimated from Eq. (2). A RH-





dependent $\gamma$ (= (2–5)×10$^{-5}$, Eq. (3)) derived from Zheng et al.(2015a)during 2013 Beijing haze was used. This range of $\gamma$ is
also consistent with the estimated values of $\gamma$ from (1.6±0.7) to (4.5±1.1) ×10$^{-5}$ by Wang et al.(2016).
$$S_{p} = \frac{PM_{2.5} \times 10^{-6}}{4/3 \times \pi R_{p}^{3} \times \rho} \times 4\pi R_{p}^{2} \quad (2)$$
$$\gamma = \begin{cases} 2 \times 10^{-5}, \text{RH} \leq 50\,\% \\ 2 \times 10^{-5} + \frac{5 \times 10^{-5} - 2 \times 10^{-5}}{100 - 50} \times (\text{RH} - 50), \; 50\,\% \leq \text{RH} \leq 100\,\% \end{cases} \quad (3)$$

**2.4 Estimate of primary sulfate**

The primary sulfate, which is directly emitted into air, includes the sea salt source, terrigenous source and
anthropogenic source (Li et al., 2013;Faloona, 2009). The concentration of sea salt sulfate was calculated by using the
observed concentrations of $SO_4^{2-}$ and $Na^+$ and the mass ratio of ($SO_4^{2-}/Na^+$) = 0.252 in seawater (Calhoun et al., 1991). The
terrigenous sulfate was estimated using the observed concentrations of $SO_4^{2-}$ and $Ca^{2+}$ and the mass ratio of ($SO_4^{2-}/Ca^{2+}$) =
0.18 in soil (Legrand et al., 1997), where ($Ca^{2+}/Na^+$) = 0.038 in seawater was used to calculate the fraction of observed $Ca^{2+}$
from soil (Legrand and Mayewski, 1997). The anthropogenic primary sulfate is estimated as 3% of anthropogenic $SO_2$
emissions in models (Faloona, 2009;Alexander et al., 2009). Supposing all the observed concentrations of $SO_2$ and
precursors of secondary sulfate are anthropogenic, we have $n_{ap}$ = 3% × ($n_{SO2}$ + $n_{sas}$), where $n_{sas}$ = $n_{tos}$–$n_{ss}$–$n_{ts}$–$n_{ap}$ and $n_{ap}$, $n_{sas}$,
$n_{tos}$, $n_{ss}$ and $n_{ts}$ is the molar concentrations of anthropogenic primary sulfate (ap), secondary sulfate (sas), total sulfate (tos),
sea salt sulfate (ss) and terrigenous sulfate (ts). The estimated concentration of total primary sulfate is the sum of primary
sulfate from all these sources.

**2.5 Estimate of sulfate production rate from OH oxidation in the gas-phase**

The sulfate production rate from OH oxidation in the gas-phase ($P_{SO2+OH}$) can be expressed as:
$$P_{SO_2+OH} = \frac{3600 \times 96 \times R_{SO_2+OH}}{RT} \quad (4)$$
where$P_{SO2+OH}$ is in unit of µg m$^{-3}$ h$^{-1}$, 3600, 96, $R$ and $T$ is the same as Eq. (1).$R_{SO2+OH}$ is the chemical reaction rate (ppb s$^{-1}$),
calculated as shown in Table S1 and S2.

**2.6 Estimate of in-cloud sulfate production rate**

The main in-cloud sulfateformation pathways considered here include S(IV) oxidation by $H_2O_2$, $O_3$, $NO_2$(Wang et al.,
2016) and $O_2$ via a radical chain mechanism initiated by TMIs(Alexander et al., 2009). Their chemical reaction rate
expressions ($R_{S(IV)+oxi}$) and rate constants ($k$) are summarized in Table S3. The rate of in-cloud sulfate production by a certain
oxidant ($P_{cloud, S(IV)+oxi}$) can be expressed as(Seinfeld and Pandis, 2012):
$$P_{cloud,S(IV)+oxi} = 3600 \times 96 \times LWC \times R_{S(IV)+oxi} \quad (5)$$





Where $P_{\text{cloud, S(IV)+oxi}}$ is in unit of µg m$^{-3}$ h$^{-1}$, 3600 and 96 is the same as Eq. (1), and $R_{\text{S(IV)+oxi}}$ is in unit of M s$^{-1}$. Cloud liquid
water content (LWC, in unit of mg m$^{-3}$) was derived from a global reanalysis, GEOS-FP
(https://gmao.gsfc.nasa.gov/products/).

**2.7 Isotopic constrainson sulfateformation pathways**

Since S(IV) oxidation by O$_3$ and H$_2$O$_2$ are the sole sources of non-zero $\Delta^{17}O(SO_4^{2-})$(Table 1), the relative importance of
different sulfate formation pathways can be calculated as follows (Alexander et al., 2012):
$\Delta^{17}O_{\text{obs}} = \left(6.5 \times f_{\text{S(IV)+O}_3}\right) + \left(0.7 \times f_{\text{S(IV)+H}_2\text{O}_2}\right) + \left(0 \times f_{\text{zero} -\Delta^{17}O}\right)$   (6)
where $f_{\text{S(IV)+O3}}$ and $f_{\text{S(IV)+H2O2}}$ are the fractional contributions of S(IV) oxidation by O$_3$ and H$_2$O$_2$ oxidation to total sulfate
production, respectively, and $f_{\text{zero-}\Delta17O}$ represents the fractional contribution of sulfate with zero-$\Delta^{17}$O processes such as
primary sulfate, secondary sulfate formed via OH oxidation, NO$_2$ oxidation, and O$_2$ oxidation via a radical chain reaction
initiated by TMIs in cloudsor due to the speciality of interfacial water on acidic microdroplets. By definition, we have
$f_{\text{S(IV)+O3}} + f_{\text{S(IV)+H2O2}} + f_{\text{zero-}\Delta17O} = 100\%$.
In addition, as sulfate with non-zero $\Delta^{17}O(SO_4^{2-})$ is produced either via in-cloud reactions or via heterogeneous
reactions or both, Eq. (6) can also be written as follows:
$\Delta^{17}O_{\text{obs}} = f_{\text{het}} \times \Delta^{17}O_{\text{het}} + f_{\text{cloud}} \times \Delta^{17}O_{\text{cloud}} + f_{\text{SO}_2+\text{OH}} \times \Delta^{17}O_{\text{SO}_2+\text{OH}} + f_{\text{p}} \times \Delta^{17}O_{\text{p}}$   (7)
Where $f_{\text{het}}, f_{\text{cloud}}, f_{\text{SO2+OH}}$ and $f_{\text{p}}$respectivelyrepresents the fractional contribution of heterogeneous sulfate production, in-cloud
sulfate production, gas-phase sulfate production and primary sulfate to the observed sulfate.$\Delta^{17}O_{\text{het}}$, $\Delta^{17}O_{\text{cloud}}$, $\Delta^{17}O_{\text{SO2+OH}}$
and $\Delta^{17}O_{\text{p}}$ respectively represents $\Delta^{17}$O of corresponding sulfate produced via above pathways. Both $\Delta^{17}O_{\text{SO2+OH}}$ and $\Delta^{17}O_{\text{p}}$ is
equal to 0‰. $\Delta^{17}O_{\text{cloud}}$ can be calculated as Eq. (8) due to thatlifetime of sulfate produced in clouds will not depend on the
specific S(IV) oxidant.
$\Delta^{17}O_{\text{cloud}} = \frac{6.5 \times P_{\text{cloud},\text{S(IV)+O}_3} + 0.7 \times P_{\text{cloud},\text{S(IV)+H}_2\text{O}_2}}{P_{\text{cloud}}}$   (8)
Where $P_{\text{cloud}}$ is the rate of total in-cloud sulfate production, which was calculated as the sum of in-cloud S(IV) oxidation by
H$_2$O$_2$, O$_3$, NO$_2$ and O$_2$initiated by TMIs.

**2.8 The prediction of aerosol water content (AWC), aerosol pH and ionic strength ($I_s$)**

AWC, aerosol pHand $I_s$ was calculated by the ISORROPIA II model, which is a thermodynamic equilibrium model for
NH$_4^+$-K$^+$-Ca$^{2+}$-Na$^+$-Mg$^{2+}$-SO$_4^{2-}$-NO$_3^-$-Cl$^-$-H$_2$O aerosols (Fountoukis and Nenes, 2007). The ISORROPIA II model can solve
forward problems in which $T$, RH and the concentrations of gas + aerosols are known (eg: NH$_3$ + NH$_4^+$), and reverse
problems in which $T$, RH and the concentrations of aerosol (but not gas) species are known. We used the forward method to
calculate AWC, aerosol pH and $I_s$ as this method has been shown to best predict aerosol pH (Hennigan et al., 2015). The
AWC, pH and $I_s$ was firstly calculated in metastable mode (assuming that bulk aerosol solution is supersaturated), which is
consistent with previous studiesabout Beijing haze (Liu et al., 2017;Guo et al., 2017). However, the work ofRood et al.(1989)



in California, USA suggested that not all aerosolsare in metastable state, even though the fractional occurrence of metastable
aerosolsincreases with increasing RH in urban sites (i.e., from near 0 at RH <round 30% to near 100 % at RH > round 80%,
roughly following Eq. (9)). So we also predicted the AWC, pH and $I_s$ in stable mode (assuming that bulk aerosols crystallize
once saturation is exceeded), which is consistent with Wang et al.(2016).The input of observed inorganic ion concentrations
and meteorological parametersare summarized in Table S4. Since gaseous $NH_3$ was not measured in our campaign, we used
the empirical equation $NH_3$ (ppb) = 0.34×$NO_X$(ppb)+0.63, derived from observations of Meng et al.(2011) in Beijing winter,
to estimate the $NH_3$ concentrations. We used $NO_2$ concentrations instead of $NO_X$ as input due to the lack of $NO_X$
observations in our study, which would give a lower end of the $NH_3$ concentrations. Given the importance of AWC for
reaction rates and the fact that ISORROPIA II underestimates AWC at low RH (Bian et al., 2014), samples with RH < 40%
are excluded from analysis (Hennigan et al., 2015). This excludes 8 out of the total 34 samples (24%), with 6 of them in
NPD. A total of 4 samples in NPD and 22 samples in PD were analysed for AWC, aerosol pH and $I_s$ using observations and
the ISORROPIA II model. Due to that the predicted $I_s$ is high ($I_s$> 10 M, Table S4), which suggests aerosol water is non-ideal,
the influence of $I_s$ on reaction rate constants (Table S3) and effective Henry's law constants (Table S5) is taken into
considerationwhen the influence is known.
$$MF = \begin{cases} 0, & RH < 30\,\% \\ -0.024 \times RH^2 + 4.18 \times RH - 89.13, & 30\,\% \leq RH \leq 80\,\% \\ 100\,\%, & 80\,\% < RH \leq 100\,\% \end{cases} \quad (9)$$
where MF (in %) is the fraction of metastable aerosols to total aerosols.

## 2.9 Estimate of aqueous concentrations of trace species

The aqueous concentrations of $SO_2$, $O_3$, $H_2O_2$ and $NO_2$ were calculated as described in Table S5. The determination of
in-cloud concentrations of TMIs (here only Fe(III) and Mn(II) (Alexander et al., 2009)) is described below.
The concentration of soluble Fe(III) follows Eqs. (10)–(13)(Liu and Millero, 1999):
$\log_{10}[\text{Fe(III)}] = \log_{10} K^*_{\text{Fe(OH)}_3} + 3 \times \log_{10}[\text{H}^+] + \log_{10}(1 + \beta_1^*[\text{H}^+]^{-1} + \beta_2^*[\text{H}^+]^{-2})$  (10)
where
$\log_{10} K^*_{\text{Fe(OH)}_3} = -13.486 - 0.1856 \times I_s^{0.5} + 0.3073 \times I_s + 5254/T$  (11)
$\log_{10} \beta_1^* = 2.517 - 0.8885 \times I_s^{0.5} + 0.2139 \times I_s - 1320/T$ (12)
$\log_{10} \beta_2^* = 0.4511 - 0.3305 \times I_s^{0.5} - 1996/T$ (13)
and [Fe(III)] is the aqueous concentration of Fe(III) in unit of M, $T$ is temperature in unit of K, and $I_s$ is ionic strength in unit
of M, $K^*_{\text{Fe(OH)}3}$ is the solubility product constant of Fe(OH)$_3$, and $\beta^*_1$ and $\beta^*_2$ is respectively first-order and second-order
cumulative hydrolysis constants of Fe$^{3+}$.
Our calculation suggests in-cloud [Fe(III)] was in the range of 0.6 to 6.1 μM with a mean of (2.6±1.8) μM, which is
similar to the observed values in NCP(Guo et al., 2012;Shen et al., 2012). The concentration of soluble Mn(II) in cloud water



was set to be 1 μM in the present study, which is the general value observed in cloud water in NCP (Guo et al., 2012;Shen et
al., 2012).

**2.10 Estimate of sulfate production rate in aerosol water**

The reaction rate expressions, rate constants ($k$) and the influence of $I_s$ on $k$ for sulfate production in aerosol water are
summarized in Table S3. The overall rates for S(IV) oxidation in aerosol water depend not only on chemical reaction rates
(Table S3) but also on the mass transport limitations. A standard resistance model was used to estimate effects of mass
transport following the work ofCheng et al.(2016):
$\dfrac{1}{R_{\text{H,S(IV)}+\text{oxi}}} = \dfrac{1}{R_{\text{S(IV)}+\text{oxi}}} + \dfrac{1}{J_{\text{aq,lim}}}$   (14)
where $R_{\text{H, S(IV)}+\text{oxi}}$ is the overall reaction rate for S(IV) oxidation by a certain oxidant (oxi) such as $O_3$, $H_2O_2$, $NO_2$ and $O_2$on
acidic microdroplets (M s$^{-1}$), $R_{\text{S(IV)}+\text{oxi}}$ is the chemical reaction rate (M s$^{-1}$) and $J_{\text{aq, lim}}$ is the rate limited by mass transfer from
the gas to the aqueous phase (M s$^{-1}$). $R_{\text{S(IV)}+\text{oxi}}$ was calculated as described in Table S3. The limiting mass transfer $J_{\text{aq, lim}}$ was
calculated by Eqs.(15) and (16).
$J_{\text{aq,lim}} = \min\{J_{\text{aq}}(\text{SO}_2), J_{\text{aq}}(\text{oxi})\}$   (15)
$J_{\text{aq}}(\text{X}) = k_{\text{MT}}(\text{X}) \times [\text{X(aq)}]$   (16)
where X = $SO_2$, $O_3$, $H_2O_2$ or $NO_2$ and $k_{\text{MT}}$ (s$^{-1}$) is the mass transfer rate coefficient and was calculated as Eq. (17)(Cheng et
al., 2016;Seinfeld and Pandis, 2012):
$k_{\text{MT}}(\text{X}) = \left[\dfrac{R_p^2}{3D_g} + \dfrac{4R_p}{3\alpha v}\right]^{-1}$   (17)
Where $R_p$, $D_g$ and $v$ are the same as Eq.(1). The $\alpha$ used in our calculation is respectively 0.11 for $SO_2$, 0.23 for $H_2O_2$, 2.0×10$^{-}$
$^3$ for $O_3$ and 2.0×10$^{-4}$ for $NO_2$(Seinfeld and Pandis, 2012;Jacob, 2000). The term on the left hand side of Eq.(17) is the gas-
phase diffusion limitation while the term on the right hand side of Eq. (17) is the interfacial mass transport limitation. $k_{\text{MT}}$
was limited by interfacial mass transport limitation in our study.
The rate of heterogeneous sulfate production by a certain oxidant ($P_{\text{het, S(IV)}+\text{oxi}}$) in aerosol water can be expressed as:
$P_{\text{het,S(IV)}+\text{oxi}} = 3600 \times 96 \times \text{AWC} \times R_{\text{H,S(IV)}+\text{oxi}}$   (18)
Where $P_{\text{het, S(IV)}+\text{oxi}}$ is in the unit of μg m$^{-3}$ h$^{-1}$, 3600 and 96 is the same as Eq. (1). AWC is in the unit of mg m$^{-3}$ and $R_{\text{H,}}$
$_{\text{S(IV)}+\text{oxi}}$ is in the unit of M s$^{-1}$, For $R_{\text{H, S(IV)}+\text{oxi}}$, our calculation suggested that the role of mass transport limitationsfor S(IV)
oxidation by $NO_2$ was significant at high pH values.



## 3 Results and Discussion

### 3.1 Characteristics of haze events in Beijing

Figure 1ashows the temporal evolution of concentrations of $PM_{2.5}$ and $SO_4^{2-}$ during our sampling period. The 12h-averaged $PM_{2.5}$ concentrations ranged from 16 to 323 µg m$^{-3}$ with a mean of $(141\pm88$ $(1\sigma))$ µg m$^{-3}$. In comparison, the Grade II of the Chinese National Ambient Air Quality Standard of daily $PM_{2.5}$ is 75 µg m$^{-3}$. The $SO_4^{2-}$ concentrations varied from 1.5 to 56.4 µg m$^{-3}$ with a mean of $(21.2\pm15.4)$ µg m$^{-3}$. As shown in Fig. 1a, $SO_4^{2-}$ concentrations presented a similar temporal trend as $PM_{2.5}$ concentrations, i.e., increased from a mean of $(3.9\pm1.8)$ µg m$^{-3}$ in non-polluted days (NPD, $PM_{2.5}<$ 75 µg m$^{-3}$) to $(28.4\pm12.5)$ µg m$^{-3}$ in polluted days (PD, $PM_{2.5}\geq$ 75 µg m$^{-3}$). The fraction of $SO_4^{2-}$ to $PM_{2.5}$ mass concentration ranged from 8–25%, and increased from a mean of $(11\pm2)$% in NPD to $(15\pm5)$% in PD. The sulfur oxidation ratio (SOR = $nSO_4^{2-}/(nSO_4^{2-}+nSO_2)$, where $nSO_4^{2-}$ and $nSO_2$ represents the molar concentration of $SO_4^{2-}$ and $SO_2$, respectively), a proxy for secondary sulfate formation (Sun et al., 2006), also increased rapidly with $PM_{2.5}$ levels, from a mean of $(0.12\pm0.04)$ in NPD to $(0.41\pm0.17)$ in PD (Fig. 1b).

Observed $\Delta^{17}O(SO_4^{2-})$ $(\Delta^{17}O_{obs})$ ranged from 0.1‰ to 1.6‰ with a mean of $(0.9\pm0.3)$‰ (Fig. 1b). The highest $\Delta^{17}O_{obs} =$ 1.6‰ occurred during PD of Case II in October 2014 while the lowest $\Delta^{17}O_{obs} = 0.1$‰ occurred during PD of Case IV in December 2014. $\Delta^{17}O_{obs}$ reported here is similar in magnitude to previous observations of $\Delta^{17}O(SO_4^{2-})$ in aerosols and rainwater collected at other mid-latitude sites (Table S6). The overall $\Delta^{17}O_{obs}$ levels during our entire sampling time are similar for NPD and PD, being $(0.9\pm0.1)$‰ and $(0.9\pm0.4)$‰, respectively. However, the NPD to PD difference of $\Delta^{17}O_{obs}$ can be case-dependent. For Case I and II in October 2014, $\Delta^{17}O_{obs}$ increased from NPD to PD, while the opposite trend is observed for Case III to V in November 2014 to January 2015 (Fig. 2a). These $\Delta^{17}O_{obs}$ variations are generally similar to variability in concentrations of observed $O_3$ and calculated $H_2O_2$(Fig. 2b&c), which is consistent with the fact that $O_3$ and $H_2O_2$ are the sole sources of non-zero $\Delta^{17}O(SO_4^{2-})$ (Table 1).

### 3.2 Direct estimate of sulfate formation pathways based on $\Delta^{17}O_{obs}$

The fact that $\Delta^{17}O_{obs}$ falls out of the range of any single reaction pathway suggests that sulfate in Beijing haze must be produced by multiple reactions. Figure3 shows the calculated possible fractional contributions of each formation pathway $(f_{S(IV)+H2O2}, f_{S(IV)+O3},$ and $f_{zero-\Delta17O})$ for each sample using Eq. (6). On average over all samples collected, $f_{S(IV)+O3} = 4–13\%$, $f_{S(IV)+H2O2} = 0–88\%$, and $f_{zero-\Delta17O} = 8–87\%$. For samples during PD of Case IV in December 2014 with the three lowest $\Delta^{17}O_{obs}$ values (Fig. 1b), $f_{zero-\Delta17O}$ was respectively in the range of 57–95%, 86–98% and 57–95%, corresponding to $f_{S(IV)+H2O2}$ being in the range of 0–43 %, 0–14 % and 0–43% respectively, which suggests zero-$\Delta^{17}O$ pathways clearly dominated sulfate formation during PD of Case IV. However, for other samples, the maximum possible $f_{S(IV)+H2O2}$ ranged from 71 to 100% with a mean of $(93\pm7)$% while the maximum possible $f_{zero-\Delta17O}$ was 75 to 92% with a mean of $(86\pm4)$%, implying that sulfate formation during these sampling periods were dominated by $H_2O_2$ oxidation and/or zero-$\Delta^{17}O$ pathways.



### 3.3 Chemical kinetic calculations with the constraint of $\Delta^{17}O_{obs}$

The good correlation between RH and SOR in Fig. 4a (r = 0.76, p < 0.01) suggests heterogeneous reactions played an important role in sulfate formation. Our calculations show that overall heterogeneous sulfate production ($P_{het}$, see Sect. 2.3) presented similar trends with $PM_{2.5}$ concentrations (Fig. 4b) and increased from a mean of (0.6±0.3) μg m$^{-3}$ h$^{-1}$ in NPD to (2.0±1.1) μg m$^{-3}$ h$^{-1}$ in PD during our sampling period. In comparison, Cheng et al.(2016) reported that the missing sulfate production rate required to explain the observed sulfate concentration is around 0.07 μg m$^{-3}$ h$^{-1}$ when $PM_{2.5}$< 50 μg m$^{-3}$ and around 4 μg m$^{-3}$ h$^{-1}$ when $PM_{2.5}$> 400 μg m$^{-3}$ during 2013 Beijing haze. We also calculate the contribution from primary sulfate and perform chemical kinetic calculations including $SO_2$ oxidation by OH in the gas-phase and in-cloud sulfate production (Fig. 5, see Sect. 2.4–2.6)to estimate the relative importance of heterogeneous sulfate production during haze in our sampling period. Heterogeneous reactions were found to dominate sulfate formation during PD in four out of the total five Cases with fractional contribution of 42 to 54% and a mean of (48±5)% (Fig. 5). This is consistent with Zheng et al.(2015a) who reported that about half of the observed sulfate was from heterogeneous reactionsduring 2013 Beijing haze. In contrast, we found that during PD of Case II in October 2014, heterogeneous sulfate production only accounted for 23% of total sulfate production while in-cloud sulfate production predominated total sulfate production with an estimated fraction of 68%. The predominant role of in-cloud sulfate production in PD of Case II was supported by the relative high LWC during this time period (Fig. 6a). Our calculations also suggest the in-cloud sulfate production was dominated by $H_2O_2$ oxidation throughout our sampling period (Fig. 6b), which is consistent with previous findings that $H_2O_2$ oxidation is the most important in-cloud sulfate production pathway globally (Seinfeld and Pandis, 2012) and in NCP (Shen et al., 2012). In addition, the $\Delta^{17}O$ of sulfate produced in clouds ($\Delta^{17}O_{cloud}$) was estimated to range from 0.5‰ to 0.8‰ with a mean of (0.6±0.1)‰ during our sampling period and showed similar variations with $\Delta^{17}O_{obs}$(Fig. 6c). The mean value of $\Delta^{17}O_{cloud}$calculated here is close to $\Delta^{17}O(SO_4^{2-})$ in rainwater observed in central China (0.53±0.19 ‰) (Li et al., 2013) and at Baton Rouge, USA (0.62±0.32 ‰) (Jenkins and Bao, 2006).In addition, by using Eq. (7), the$\Delta^{17}O$ of sulfate produced via heterogeneous reactions($\Delta^{17}O_{het}$) was calculated to range from 0.1 ‰ to 3.1‰ in our study.

To explore the specific mechanisms of heterogeneous oxidation of $SO_2$, we calculate aerosol parameters such as aerosol water content (AWC), pH and ionic strength ($I_s$) by using the ISORROPIA II thermodynamic model (Fountoukis and Nenes, 2007)(Fig.7, see Sect. 2.8). It was found that the assumptions about aerosol thermodynamic state (salts crystallize once saturation is exceeded, termed as "stable state" or aerosol solution is supersaturated, termed as "metastable state") significantly influence the calculated aerosol pH, but have little impact on the calculated AWC and $I_s$(Fig. 7). Calculated AWC increased with $PM_{2.5}$ concentrations, from (5.3±7.4) μg m$^{-3}$ in NPD to (63.5±54.6) μg m$^{-3}$ in PD when assuming stable state and from (9.6±6.0) μg m$^{-3}$ in NPD to (84.2±49.2) μg m$^{-3}$ in PD when assuming metastable state (Fig. 7a). Calculated $I_s$ was similar for stable and metastable assumptions, ranging from 11.3 to 51.6 M (Fig. 7b). The high $I_s$ suggested aerosol water was non-ideal and thus the influence of $I_s$ on reaction rate constants (Table S3) and effective Henry's law constants (Table S5) was taken into consideration when the influence is known. The bulk aerosol pH predicted in stable state was in



the range of 7.5 to 7.8 with a mean of (7.6±0.1), consistent with bulk aerosol pH (7.63±0.03) calculations during a haze
event in Beijing 2015 predicted by Wang et al.(2016). The bulk aerosol pH calculated assuming metastable state was in the
range of 3.4 to 7.6 with a mean of (4.7±1.1), consistent with the mean value of 4.2 calculated in metastable aerosol
assumption during severe haze in Beijing 2015–2016 by Liu et al.(2017). The calculated aerosol pH assuming metastable
state decreased with increasing $PM_{2.5}$ levels, from a mean of (6.5±1.3) in NPD to (4.4±0.6) in PD, while that assuming stable
state shows no relationship with $PM_{2.5}$ concentrations (Fig. 7c). Our measured pH of filtrate ranged from 4.6 to 8.2 with an
mean of (5.7±1.0), similar to pH of filtrate from $PM_{2.5}$ in Beijing reported by Wang et al.(2005). The measured pH of filtrate
shows similar trends with bulk aerosol pH calculated assuming metastable state (Fig. 7c), with a mean value (6.9±0.7) in
NPD and (5.1±0.6) in PD, which suggests that bulk aerosols are in metastable statewith moderate acidityin PD. This is also
consistent with our estimate that most aerosols are in metastable with a fraction of (74±17) % in PD by using Eq. (9) and our
cognition that the mixture of major acidic aerosols with minor neutralaerosols would lead to the bulk being acidic. However,
as the effective Henry's law constant of $SO_2$ at pH= 7.6(stable state) can be 3 orders magnitude higher than that at pH = 4.4
(metastable state in PD) (Seinfeld and Pandis, 2012), the high pH could render stable state aerosols, which are minorities
although, being potentiallysignificant active sites for heterogeneous sulfate production during PD.

319       The main heterogeneous sulfate formation pathways include S(IV) oxidation by $H_2O_2$, $O_3$, $NO_2$ and $O_2$initiated due to

the speciality of interfacial water on acidic microdroplets as proposed by Hung and Hoffmann(2015). Other sulfate
formation pathways such as S(IV) oxidation by $NO_3$ radical, methyl-hydrogenperoxide (MHP), peroxyacetic acid (PAA),
and hypohalous acids in aerosol water (Feingold et al., 2002;Walcek and Taylor, 1986;Chen et al., 2017) is thought to be
negligible during haze in NCP (Cheng et al., 2016), and thus is not considered here. We estimate the relative importance of
main heterogeneous sulfate formation pathways as the following procedure. Firstly, the heterogeneous sulfate production
rate via S(IV) oxidation by $H_2O_2$ ($P_{het, S(IV)+H2O2}$) was calculated with the influence of $I_s$ being considered, which has been
determined at high $I_s$in laboratories. Then its fractional contribution ($f_{het, S(IV)+H2O2}$) to overall heterogeneous sulfate
production ($P_{het}$) calculated using apparent γ (see Sect. 2.3) was estimated. However, large uncertainties exist in the
influence of $I_s$ on the reaction rate constant of S(IV) oxidation by $O_3$ in aerosol water (Table S3), which prevents the estimate
of its fractional contribution ($f_{het, S(IV)+O3}$) to $P_{het}$ from purely chemical kinetic calculations. Instead, $f_{het, S(IV)+O3}$was estimated
using our calculated $f_{het, S(IV)+H2O2}$ and $\Delta^{17}O_{het}$values, on the basis that $\Delta^{17}O(SO_4^{2-}) > 0$‰ originates solely from $H_2O_2$ and $O_3$
oxidation.Then zero-$\Delta^{17}O$ pathways such as S(IV) oxidation by $NO_2$in aerosol water and by $O_2$on acidic microdropletswas
estimated to bethe remaining part ($f_{het, zero-\Delta17O}$). At last, the potential importance of S(IV) oxidation by $NO_2$ in aerosol water
and $O_2$ on acidic microdroplets was discussed.

334       Calculations show that $f_{het, S(IV)+H2O2}$ was 4–6% with a mean of (5±1)% under stable aerosol assumptions, and 8–19%

with a mean of (13±4)% under metastable state assumptions for PD of all the Cases. $f_{het, S(IV)+O3}$was calculated to be 2–47%
with a mean of (22±17)% in stable state assumption and 0–47% with a mean of (21±18)% in metastable state assumption.
Correspondingly, $f_{het, zero-\Delta17O}$was the remaining 73 % (47–94%) in stable assumption, or 66% (42–81%) in metastable
assumption for PD of all the Cases (Fig. 8). Excluding PD of Case II, in which sulfate formation was predominated by in-



cloud reactions, our calculations suggest zero-$\Delta^{17}O$ pathways such as S(IV) oxidation by $NO_2$ in aerosol water and by $O_2$ on
acidic microdroplets are important for sulfate formation during Beijing haze.

341        Cheng et al. (2016) suggested that S(IV) oxidation by $NO_2$ in aerosol water could largely account for the missing sulfate

source in 2013 Beijing haze. In their study, the calculated mean aerosol pH is 5.8, while influence of $I_s$ was not taken into
account due to the lack of relevant experimental data. The calculated $P_{het, S(IV)+NO2}$ is highly sensitive to aerosol pH. In our
study, when aerosol pH decreased from (7.6±0.1) assuming stable state to (4.7±1.1) assuming metastable state, mean $P_{het,}$
$_{S(IV)+NO2}$ decreased from (6.5±7.7) μg m$^{-3}$ h$^{-1}$ to (0.01±0.02) μg m$^{-3}$ h$^{-1}$ for PD of all the Cases (Fig. 8). The former is much
larger than our estimate of overall heterogeneous production rate, $P_{het} = (2.0±1.1)$ μg m$^{-3}$ h$^{-1}$, while the latter is too small.
Moreover, the influence of $I_s$ was not considered, which, in principal, tends to increase the reaction rate constant of S(IV)
oxidation by $NO_2$(Cheng et al., 2016). The treatment of aerosols as a bulk quantity, assuming that all aerosols are either in
stable or metastable state, may lead to errors in calculating heterogeneous sulfate production rates. As stated in Sect. 2.8, not
all aerosols are in metastable state, even though the fractional occurrence of metastable aerosols increases with increasing
RH (Rood et al., 1989). It shows in Fig. 9a that the fraction of metastable aerosols to total aerosols (MF in %), estimated by
using Eq. (9), increases with PM$_{2.5}$ levels. However, $P_{het, S(IV)+NO2}$ assuming a combination of metastable and stable state as $P_{het,}$
$_{S(IV)+NO2}$ = MF×$P_{het, S(IV)+NO2, metastable}$ + (100 %–MF)×$P_{het, S(IV)+NO2, stable}$ can still increase with PM$_{2.5}$ levels and reach (0.9±0.7)
during PD of all the Cases (Fig. 9b), much higher than $P_{het, S(IV)+NO2} = (0.01±0.02)$ μg m$^{-3}$ h$^{-1}$ under sole metastable aerosol
assumption. This estimate suggested that even though the majority of aerosols may be in metastable state during PD (74 ±
17 % in our calculation), the high pH of the minority of aerosols in stable state could render S(IV) oxidation by $NO_2$  a
potentially significant pathway for heterogeneous sulfate production.

358        Since $P_{het, S(IV)+NO2}$ using calculated aerosol pH assuming metastable state was two orders of magnitude lower than $P_{het}$

during PD, we further examined S(IV) oxidation by $O_2$ on acidic microdroplets under the metastable state assumption. A
laboratory study suggested that $SO_2$ oxidation by $O_2$ on acidic microdroplets has a large aqueous-phase reaction rate constant
of 1.5×10$^6$[S(IV)] (M s$^{-1}$) at pH ≤ 3, a pH range much lower than our calculated pH values. The rate constant was shown to
decrease with increasing pH, however, no values of the rate constant at pH > 3 was reported (Hung and Hoffmann, 2015).
Fig.8b shows heterogensous sulfate production rate via S(IV) oxidation by $O_2$ on acidic microdroplets($P_{het, S(IV)+O2}$) with
AWC calculated assuming metastable state and the aqueous-phase rate constant for pH≤ 3 being used. The estimated $P_{het,}$
$_{S(IV)+O2}$ was 1520.5 to 130359.1 μg m$^{-3}$ h$^{-1}$ with a mean of (25166.8±27266.3) μg m$^{-3}$ h$^{-1}$ during PD of all Cases, which is
four order of magnitude larger than $P_{het}$. This value should be an overestimate due to our calculated bulk aerosol pH
predicted in metastable state being (4.4±0.6) during PD. However, some fraction of aerosols could have pH ≤ 3 due to the
Kelvin effect (Hung and Hoffmann, 2015) to render S(IV) oxidation by $O_2$ on acidic microdroplets a potentially important
pathway for heterogeneous sulfate production.




## 4 Conclusions

Our study suggests that both heterogeneous reactions (Case I and III-V) and in-cloud reactions (Case II) can dominate sulfate formation during Beijing haze. The $\Delta^{17}O$-constrained calculation shows that the heterogeneous sulfate production during haze events in our observation was mainly (66 to 73% on average) from reactions that result in sulfate with $\Delta^{17}O=0‰$, i.e., S(IV) oxidation by $NO_2$ and/or S(IV) oxidation by $O_2$ on acidic microdroplets. S(IV) oxidation by $H_2O_2$ and $O_3$ accounted for the rest (27 to 34%) of heterogeneous sulfate production. However, given the large difference in predicted aerosol pH assuming metastable aerosol state and stable aerosol state (pH = 7.6±0.1 and 4.7±1.1, respectively) and the strong dependence of S(IV) + $NO_2$ and S(IV) + $O_2$ on aerosol pH, we cannot quantify the relative importance of these two pathways for heterogeneous sulfate production. S(IV) + $NO_2$ in aerosol water can be the dominant pathway when aerosols are in stable state with pH = 7.6±0.1, while S(IV) + $O_2$ can take over providing that highly acidic aerosols (pH≤ 3) exist.To distinguish which of these two mechanisms is more important for sulfate formation during Beijing haze, the heterogeneity of aerosol state and pH should be considered in future studies.

## Supplementary Materials

**Table S1.** Reaction rate expression and constant for $SO_2$ oxidation by OH in the gas-phase.

**Table S2.** The daytime average OH concentration.

**Table S3.** Aqueous-phase reaction rate expressions, rate constants ($k$) and influence of ionic strength ($I_s$) on $k$ for sulfate production in aerosol and cloud water.

**Table S4.** The input and output of ISORROPIA II model.

**Table S5.** Calculations of aqueous-phase concentrations, equilibrium constants and influence of ionic strength.

**Table S6.** Observed $\Delta^{17}O(SO_4^{2-})$ in aerosols or rainwater in mid-latitude areas.

## Data availability

All data needed to draw the conclusions in the present study are shown in this paper and/or the Supplementary Materials. For additional data related to this study, please contact the corresponding author (zqxie@ustc.edu.cn).

## Author contributions

Z.Q.X. conceived and led the study. P.Z.H., X.Y.C, S.D.F., H.C.Z., H. K. performed the field experiments and aerosol chemical composition measurements. P.Z.H. conducted oxygen isotope measurements supervised by B.A. and L.G.. P.Z.H., B.A., Z.Q.X., L.G., H.S. and Y.F.C. interpreted the data. G.J.Z. involved the oxidation pathway calculation. C.L. contributed





to the field observation. P.Z.H. wrote the manuscript with B.A., Z.Q.X. and L.G. inputs and revision. All authors involved the discussion and revision.

**Competing interests**

The authors declare no competing interests.

**Acknowledgments**

We thank A. J. Schauer and Q. J. Chen at the University of Washington for help with isotope ratio measurements. Z.Q. Xie acknowledges support from National Key Project of MOST (2016YFC0203302), NSFC (91544013), and the Key Project of CAS (KJZD-EW-TZ-G06-01). B. Alexander acknowledges support from NSF AGS 1644998. H. Su acknowledges support from National Key Project of MOST (2017YFC0210104) and NSFC (91644218).

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

**Figures and Tables**

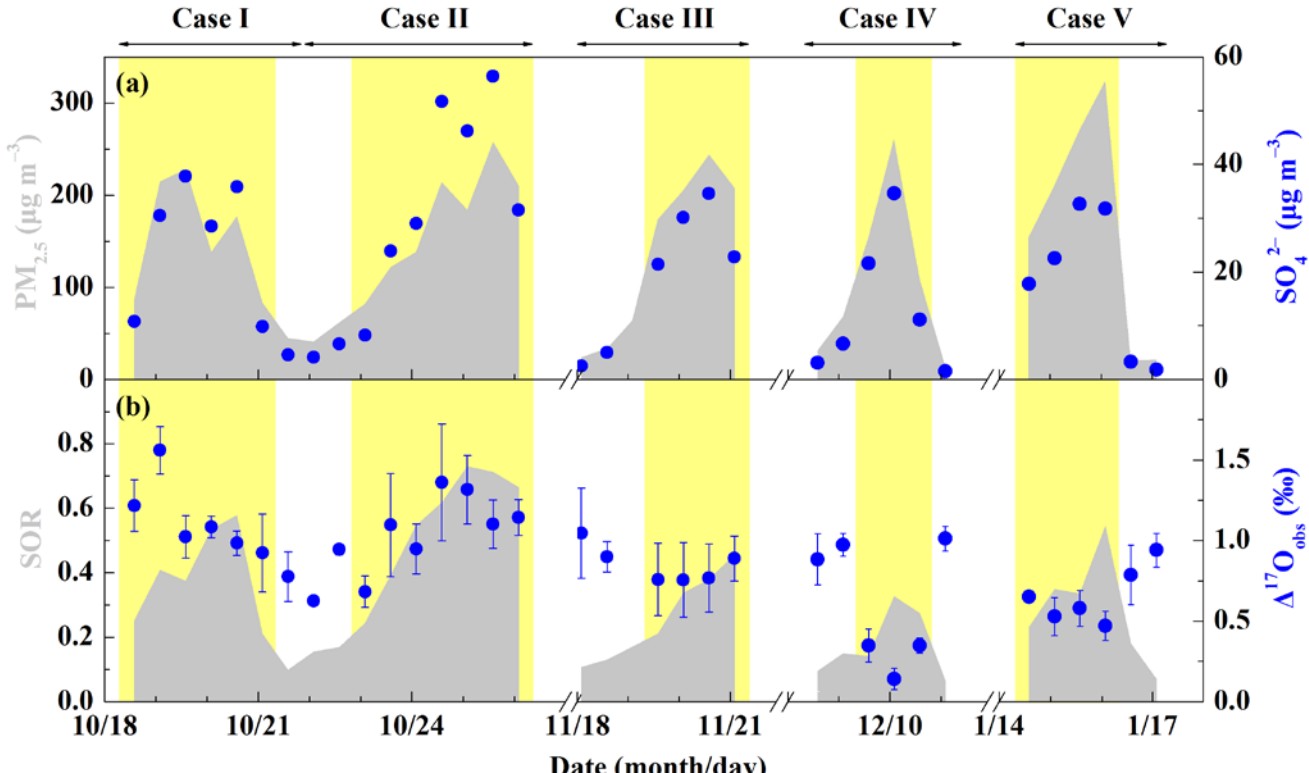


**Figure 1.** Characteristics of haze events in Beijing (October 2014–January 2015). **(a)** Temporal evolution of PM$_{2.5}$ and SO$_4^{2-}$
concentrations. **(b)**Temporal evolution of sulfur oxidation ratio (SOR = nSO$_4^{2-}$/(nSO$_4^{2-}$+nSO$_2$), n represents the molar
concentration) and observed $\Delta^{17}$O(SO$_4^{2-}$) ($\Delta^{17}$O$_{obs}$). The error bar of $\Delta^{17}$O$_{obs}$in (b) is ±1σ of replicate measurements (n = 2–4)



of each sample. The light yellow shaded area indicates polluted days (PD, $PM_{2.5} \geq 75$ µg m$^{-3}$). Data used here are 12h-
averaged values, corresponding with filter samples.

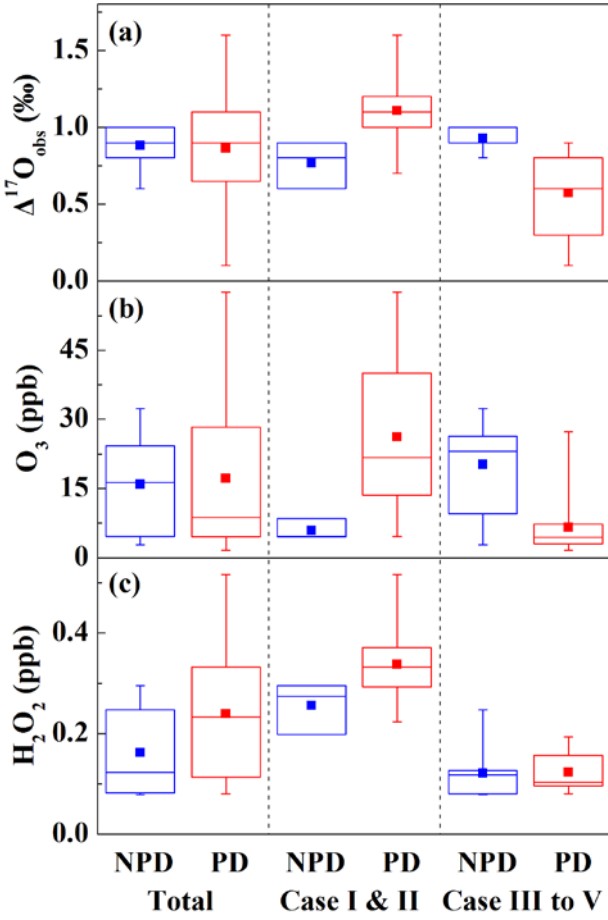


**Figure 2.** The box chart of observed $\Delta^{17}O(SO_4^{2-})$ ($\Delta^{17}O_{obs}$, **a**) and concentrations of observed $O_3$(**b**) and calculated $H_2O_2$(**c**)
in non-polluted days (NPD, $PM_{2.5} < 75$ µg m$^{-3}$) and polluted days (PD, $PM_{2.5} \geq 75$ µg m$^{-3}$). The box line from bottom to top is
respectively percentile of 25%, 50% and 75%, the whisker from bottom to top is respectively the minimum and the
maximum, and the square is mean value.



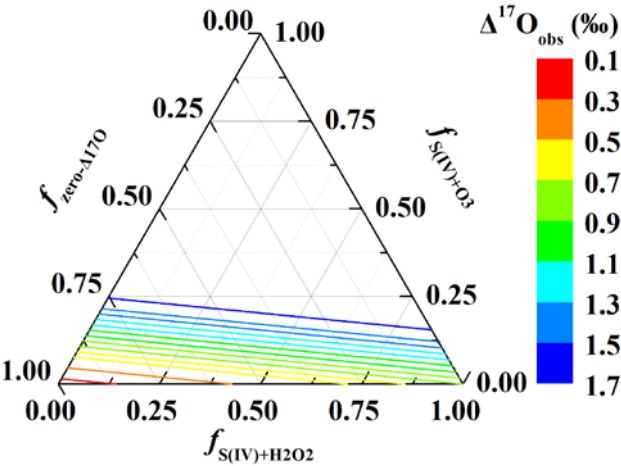


**Figure 3.** Ternary diagram of possible fractional contribution of different pathways to total sulfate production directly estimated from $\Delta^{17}O_{obs}$. The colored lines are contour lines of $\Delta^{17}O_{obs}$, representing possible fractional contribution of sulfate formation via $O_3$ ($f_{S(IV)+O3}$) and $H_2O_2$ ($f_{S(IV)+H2O2}$) oxidation or zero-$\Delta^{17}O$ processes ($f_{zero-\Delta17O}$) such as primary sulfate, secondary sulfate formed via OH oxidation, $NO_2$ oxidation and $O_2$ oxidation initiated by TMIs in clouds or due to the specialty of interfacial water on acidic microdroplets.

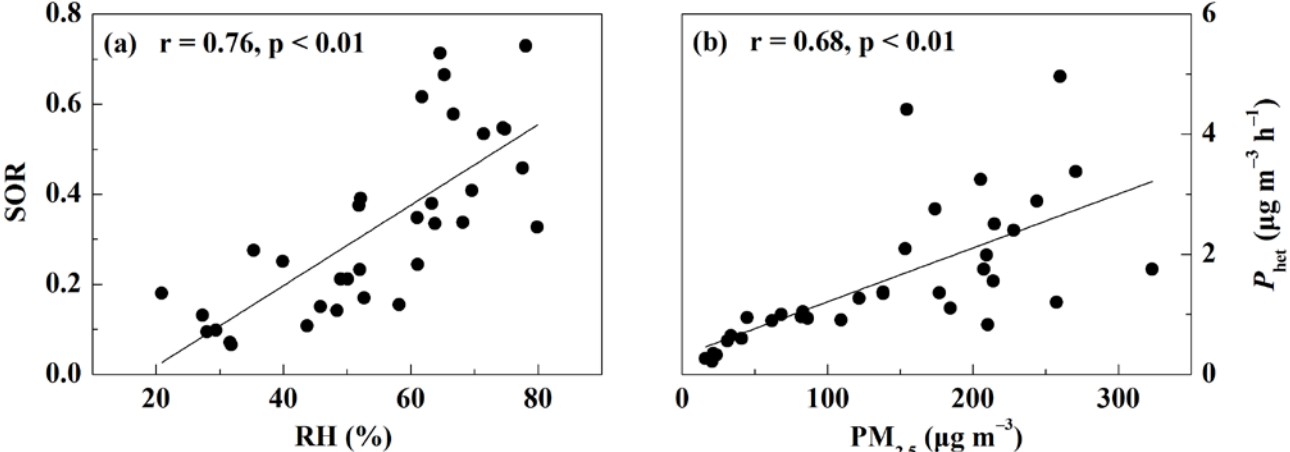

553

**Figure 4.** The relationship between RH and SOR **(a)** and relationship between $PM_{2.5}$ concentrations and rate of overall heterogeneous sulfate production ($P_{het}$, **b**). The black lines are linear least-squares fitting lines.





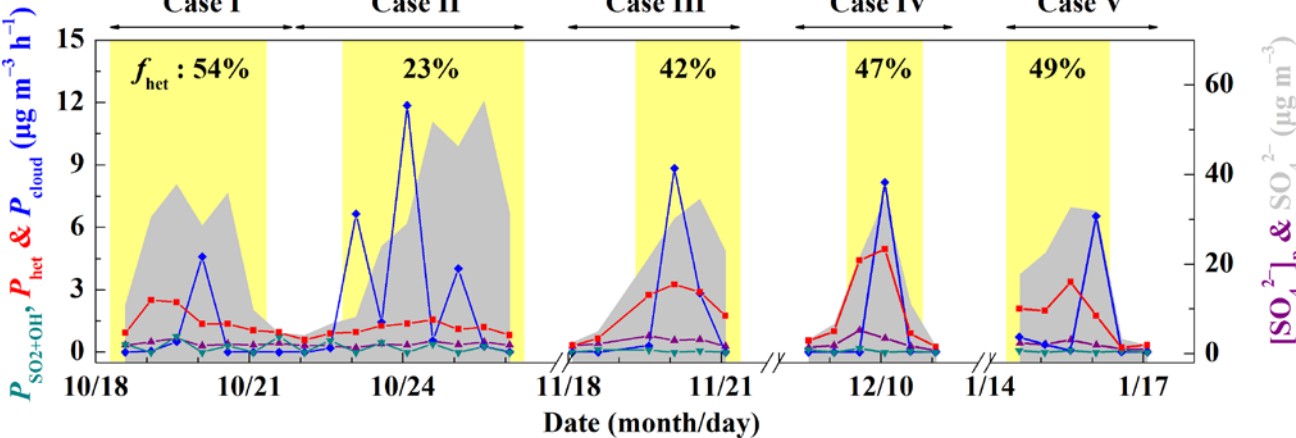

**Figure5.** Estimate of different sulfate production pathways. Time series of estimated sulfate production rate via OH oxidation in the gas-phase ($P_{SO2+OH}$), overall heterogeneous reactions on aerosols ($P_{het}$) and in-cloud reactions ($P_{cloud}$) and concentrations of primary sulfate ($[SO_4^{2-}]_p$) and observed sulfate. $f_{het}$ represents the fraction of overall heterogeneous sulfate production to total sulfate production during PD of each Case. The light yellow shaded area indicates polluted days (PD, $PM_{2.5} \geq 75\ \mu g\ m^{-3}$). Data used here are 12h-averaged values, corresponding with filter samples.





562

**Figure 6.** Temporal evolution of cloud liquid water content (LWC, **a**), in-cloud sulfate production rate via S(IV) oxidation
by $H_2O_2$, $O_3$, $NO_2$ and $O_2$ initiated by TMIs (denoted as $P_{cloud, S(IV)+H2O2}$, $P_{cloud, S(IV)+O3}$, $P_{cloud, S(IV)+NO2}$ and $P_{cloud, S(IV)+O2}$,
respectively, **b**) and estimated $\Delta^{17}O$ of sulfate produced in clouds ($\Delta^{17}O_{cloud}$, **c**). The light yellow shaded area indicates
polluted days (PD, $PM_{2.5} \geq 75$ μg m$^{-3}$). Data used here are 12h-averaged values, corresponding with filter samples.



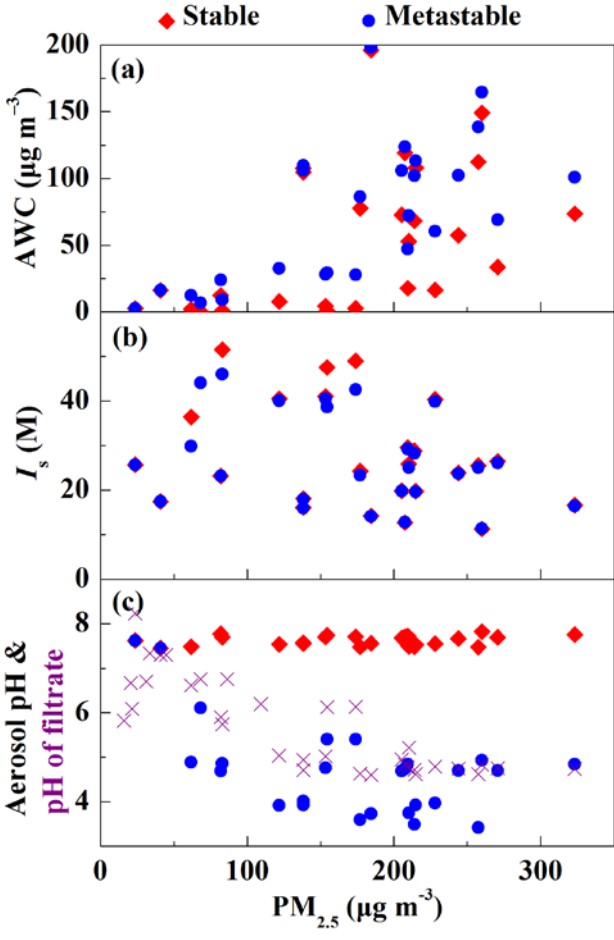

567

**Figure 7.** Aerosol parameters during Beijing haze. The aerosol water content (AWC, **a**), ionic strength ($I_s$, **b**) and aerosol pH

(**c**) was predicted by ISORROPIA II assuming stable aerosol state and metastable aerosol state. The pH of filtrate was

measured by an ion activity meter.



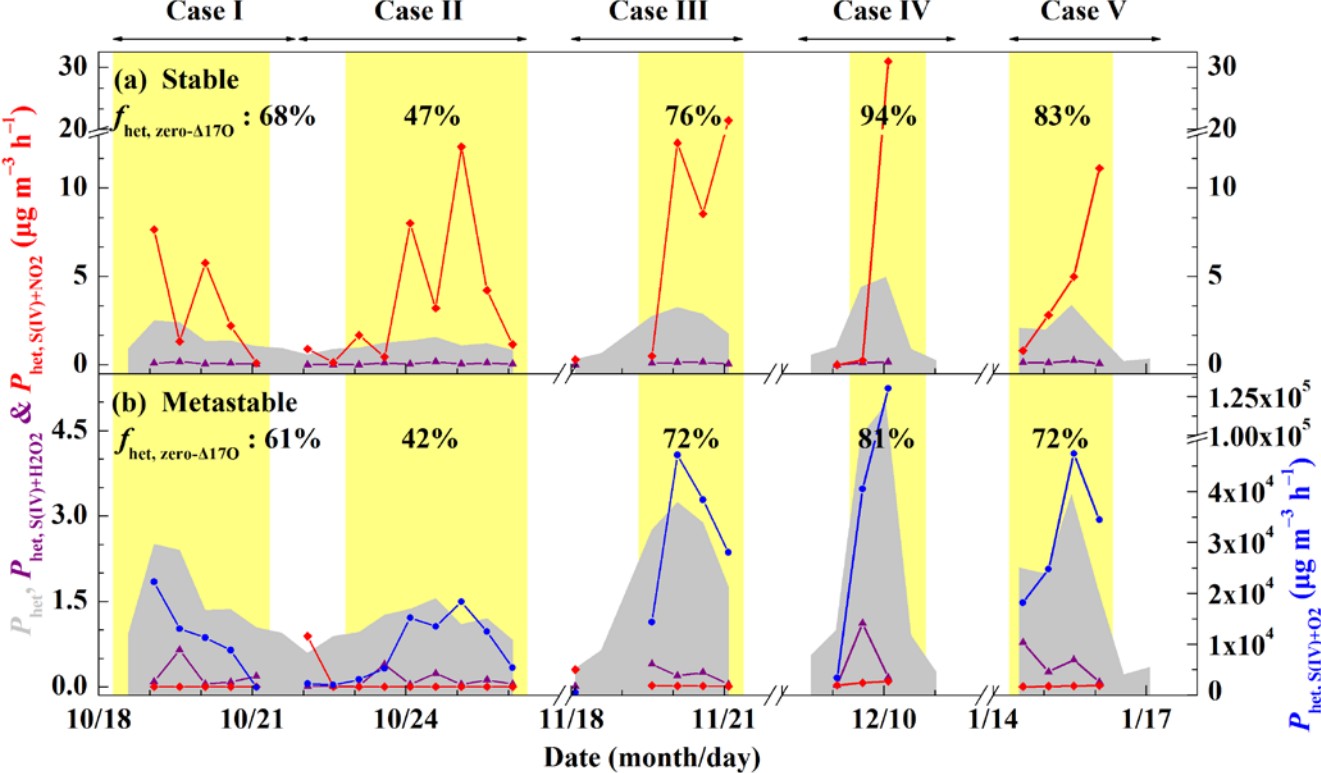

571

**Figure8.** Estimate of heterogeneous sulfate production pathways. Time series of overall heterogeneous sulfate production rate ($P_{het}$), heterogeneous sulfate production rate in aerosol water via $H_2O_2$ ($P_{het, S(IV)+H2O2}$) and $NO_2$ ($P_{het, S(IV)+NO2}$) under stable **(a)** and metastable **(b)** aerosol assumption. $P_{het, S(IV)+O2}$ in (b) represents heterogeneous sulfate production rate via $SO_2$ oxidation by $O_2$ via a radical chain mechanisminitiated due to the specialty of interfacial water on acidic microdroplets. $f_{het, zero-\Delta 17O}$ represents the fraction of heterogeneous reactions that result in sulfate with zero-$\Delta^{17}O$, such as S(IV) oxidation by $NO_2$ and $O_2$, to the overall heterogeneous sulfate production during PD of each Case with the constraint of $\Delta^{17}O(SO_4^{2-})$ (see the main text for details). In calculating $P_{het, S(IV)+H2O2}$, the influence of $I_s$ was considered. In calculating $P_{het, S(IV)+NO2}$, and $P_{het, S(IV)+O2}$the influence of $I_s$ was not considered due to the lack of experimental data about the influence of $I_s$. $P_{het, S(IV)+O2}$ was calculated using the aqueous-phase rate constant for pH $\leq$ 3 due to the lack of rate constant information at pH > 3. The light yellow shaded area indicates polluted days (PD, $PM_{2.5} \geq$ 75 µg m$^{-3}$). Data used here are 12h-averaged values, corresponding with filter samples.



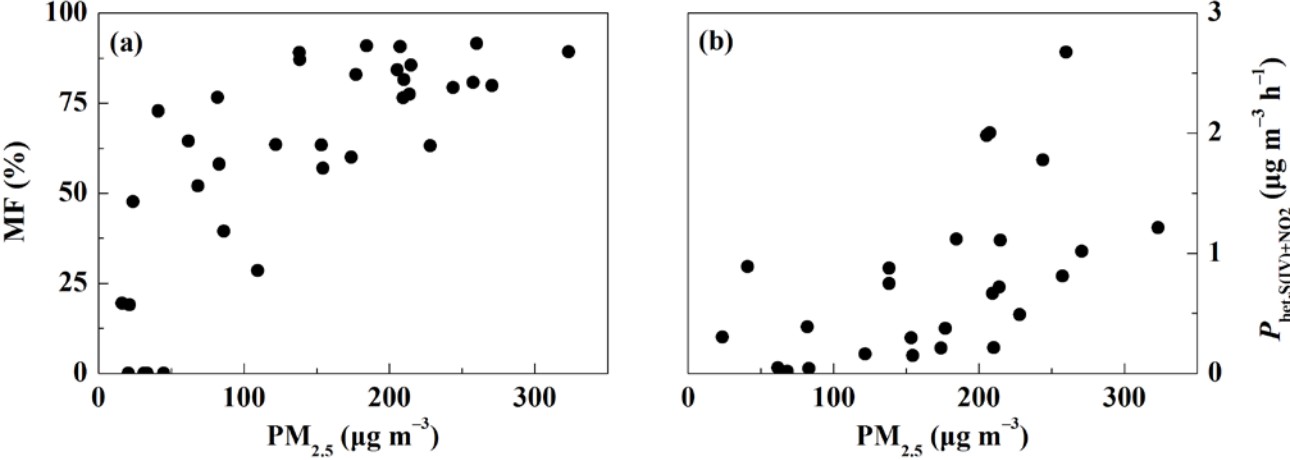

583

**Figure 9.** The estimated fraction of metastable aerosol to total aerosol (MF, **a**) using Eq. (9) and heterogeneous sulfate production rate from S(IV) oxidation by $NO_2$ assuming a combination of metastable and stable state ($P_{het, S(IV)+NO2}$, **b**) as $P_{het,}$ $_{S(IV)+NO2} = MF \times P_{het, S(IV)+NO2, metastable} + (100\% - MF) \times P_{het, S(IV)+NO2, stable}$.

**Table 1.** Sulfate isotope assumptions.

| Sulfate formation pathways | $\Delta^{17}O(SO_4^{2-})$ (‰) |
|---|---|
| $SO_2 + OH$ | 0 |
| $S(IV) + H_2O_2$ | 0.7 |
| $S(IV) + O_3$ | 6.5 |
| $S(IV) + NO_2$ | 0 |
| $S(IV) + O_2$ | 0 |
| Primary sulfate | 0 |

589