# Peer review of "Isotopic constraints on heterogeneous sulfate production in Beijing haze"

_Atmospheric Chemistry and Physics, 2017_

## Referee Comment (RC1) · Anonymous Referee #1 · 11 Dec 2017

In the manuscript, the study reported the first observations of PM2.5 $\Delta17O(SO_4^{2-})$ during haze events from October 2014 to January 2015 in Beijing, and use them to quantify the relative importance of different sulfate formation pathways, which is quite interesting and significant. But there are some mistakes and problem:

1. There are some typesetting in the manuscript: such as: (1) line 1: should be sulfate production; (2) line 25: formationwith should be formation with;(3) line 30: should be NO2 in; (4) line 84: should be round 20m; (5) line 84:should be round 60m; line 243: should be Figure 1a shows 2. Sampling site locate at a country-site (is not downtown), round 60 km northeast of downtown, and so that it is not perfect to know isotopic constraints on heterogeneous sulfate production in Beijing haze, the reason is that downtown has more information about the automobile exhaust emission, enterprise emission, and resident emission and so on. Furthermore, the sampling site is close to Yanxi lake, so that it has been probably affected by cloud liquid water content; 3. Method: As we know, these data are likely contaminated to various degrees by occluded nitrate isotope signal in the samples, so that we had better remove nitrate during the preparing the Ag2SO4, but the manuscript has not removed the nitrate, which will effect on the sulfur isotope composition results; 4. The manuscript should introduce name of laboratory, precision of the machine.

---

## Referee Comment (RC2) · M. Lin (Referee) · 11 Dec 2017

This manuscript presents a new theoretical framework for quantifying heterogeneous sulfate production pathways in Beijing haze using field-based measurements of mass independent compositions of oxygen isotopes (D17O) in sulfates. The dataset is the first measurement of D17O(SO4) in fine particles in the megacity Beijing. In addition, a combination of metastable and stable states was proposed to calculate the aerosol acidity, which plays a predominant role in the relative contributions of O2 and NO2 oxidation pathways.

The sulfate formation in Beijing haze is a subject of intense scrutiny in recent years in atmospheric chemistry community, and the use of triple oxygen isotopic analysis for

such quantification is a large step forward in this field. The isotopic measurements made in UW are of high quality as usual. Some uncertainties exist in calculation and deserve further assessment, but given that this is a first investigation and the manuscript is thorough and interesting, I strongly recommend publication in ACP after considering the following comments and suggestions.

Major comments:

1. In this manuscript, there is a big assumption that the D17O of sulfates produced by the NO2 oxidation is zero. I think the authors are probably right, but there remain uncertainties because the NO2 oxidation mechanism has not yet been defined. In the introduction, the authors cited the work of Shen and Rochelle (1998) who proposed a radical chain reaction. In this case, I agree that the sulfate product is normal. In the other work cited by the author (He et al., 2014), it was proposed that oxygen is a key oxidant and oxygen atom transfer from O2 to SO4 via NO2. The conclusion made by He et al. (2014) is based on a set of laboratory experiments, in which sulfates would not produce without O2. However, in their experiments, the role of O3 was not examined. In the ambient atmosphere (especially in urban areas), the reaction NO+O3->NO2+O2 cannot be ignored. Although it was argued that O3 was not important in Beijing haze because of its low mixing ratio, is it possible that the low O3 mixing ratio is a result of en-hanced NO+O3->NO2+O2 reaction (aka "titration effect")? As shown in many studies (e.g., Xu et al., 2011; Zhang et al., 2015), ozone mixing ratios in the urban region are lower than the surrounding rural region (probably in part due to large local emissions of fresh NO in the urban region). In this case, the oxygen anomaly in ozone molecules would lead to positive D17O values in sulfates produced via the NO2 oxidation. Be-cause of the enriched 17O in ozone, a small fraction can lead to a non-zero D17O in the sulfate product. In addition, NO+HO2->OH+NO2 is also a possible pathway to transfer anomalous oxygen atoms (with the assumption that D17O in HO2 is non-zero based on D17O values in H2O2). Can the authors provide a quantitative estimation on the possible contribution of O3/HO2 to the NO2 oxidation? Because the validity of this

assumption can significantly alter the conclusion of this paper and there is no laboratory experiment to support this assumption at present, I think such discussion would make the authors' case stronger.

2. The calculation of pH is a little bit outside of my area of expertise, but I think the authors did a good job of discussing uncertainties and caveats, and the calculation of metastable/stable states proposed by the authors seems scientifically sound. My only concern is the discussion of NO2/O2 oxidation pathways (Lines 341-369). I understand the authors want to convince readers that the NO2 oxidation could be potentially important. However, as noted by the authors, the estimated production rate of O2 oxidation is ∼4 orders of magnitude greater the P(het) (lines 358-369). Therefore, a very small fraction of aerosols with pH<3 seems enough to explain the heterogeneous production rate via O2 oxidation. In this case, why do we need the NO2 oxidation? I would like to see discussion why O2 oxidation cannot explain the heterogeneous formation here and why NO2 oxidation is required. I think discussing the O2 oxidation first and then the NO2 oxidation would make this part easier to read and follow.

Specific comments:

Line 24: Please give a quantitative context here (48+-5%?). The manuscript is focused on the heterogeneous sulfate production and therefore it is important to let readers know its overall contribution.

The introduction could be better constructed. As noted by the authors, the relative importance of O2 and NO2 oxidation pathways is highly depending on pH and is difficult to constrain (lines 60-62). In lines 76-77, the authors state that the relative importance of different sulfate formation pathways is quantified in this study. So when I read this part, I thought the authors successfully solved this problem. However, this is not the case. I think the major advantage of D17O in this study is to constrain the O3 oxidation pathway in heterogeneous sulfate formation, which is of large uncertainties in previous studies. This should be highlighted. In the end of introduction, it's better to explicitly

state something like "the contributions of O3 and H2O2 oxidation in heterogeneous sulfate formation are quantified, and the roles of NO2/O2 oxidation are discussed." to prevent any overstatement.

Lines 255-256: A recent study reported one-year D17O measurements in sulfates collected from a background mountain site in East China (Lin et al., 2017). This work is closely connected to the subject of the manuscript and should be cited.

Lines 259-260 and Fig. 2: This part is not clear to me. Do the authors mean that the D17O is directly linked to the O3/H2O2 concentrations? If this is the case, scatter plots (with correlation coefficients) or time series may be clearer. I am also confused why cases I and II were grouped together. In the rest part of this manuscript (e.g., abstract and Figure 5), case II seems significantly different from other cases. And where is case V? Please clarify. In addition, it is better to use "calculated H2O2" as the y-axis title of Fig. 2c.

Line 263: What is "the range of any single reaction pathway"? It is not clear to me. I don't think this statement is exactly correct. For example, a sample with D17O ranging from 0.6 to 1 per mil could be 100% produced from the H2O2 oxidation because D17O values in H2O2 are in the range of 1.2-2 per mil. The observed D17O value is not a supportive evidence for this statement.

Lines 274-276: Why did the author look at the PM2.5 instead of sulfate concentration? I think a good correlation between P(het) and sulfate concentrations would be more convincing. From Figure 5, it seems that the variation of SO4 is more correlated to P(cloud) than P(het). The similarity of D17O(cloud) and D17O(obs) in Figure 6 also likely indicates that the contribution of P(cloud) is more dominant than P(het). If this is the case, the role of P(het) may be overstated. I would like to see a table showing the percentages of P(het), P(cloud) and P(OH) in each case.

Lines 284-286: In Figure 5, the peak of P(cloud) is at 10/24, not exactly matching the SO4 peak at 10/25. Could the authors discuss about this difference? Is it because of

a stagnant meteorological condition? Or is it possible that the P(cloud) was underestimated?

Line 371-372: Please give a quantitative context as suggested before.

Typos:

Line 84: "around"

Line 149: "sulfate formation"

Line 282: "cases"

There are many spaces missing in the manuscript. I am not going to go through all of them.

References:

He, H., Wang, Y., Ma, Q., Ma, J., Chu, B., Ji, D., Tang, G., Liu, C., Zhang, H., and Hao, J. (2014): Mineral dust and NOx promote the conversion of SO2 to sulfate in heavy pollution days, Sci. Rep., 4, 4172.

Lin, M., Biglari, S., Zhang, Z., Crocker, D., Tao, J., Su, B., Liu, L., and Thiemens, M. H. (2017): Vertically uniform formation pathways of tropospheric sulfate aerosols in East China detected from triple stable oxygen and radiogenic sulfur isotopes, Geophys. Res. Lett., 44, https://doi.org/10.1002/2017GL073637.

Shen, C. H., and Rochelle, G. T. (1998), Nitrogen dioxide absorption and sulfite oxidation in aqueous sulfite, Environ. Sci. Technol., 509 32, 1994-2003.

Xu, J., Ma, J. Z., Zhang, X. L., Xu, X. B., Xu, X. F., Lin, W. L., Wang, Y., Meng, W., and Ma, Z. Q. (2011), Measurements of ozone and its precursors in Beijing during summertime: impact of urban plumes on ozone pollution in downwind rural areas, Atmos. Chem. Phys., 11, 12241-12252, https://doi.org/10.5194/acp-11-12241-2011.

Zheng, G. J., Duan, F. K., Su, H., Ma, Y. L., Cheng, Y., Zheng, B., Zhang, Q.,

Huang, T., Kimoto, T., Chang, D., Pöschl, U., Cheng, Y. F., and He, K. B. (2015), Exploring the severe winter haze in Beijing: the impact of synoptic weather, regional transport and heterogeneous reactions, Atmos. Chem. Phys., 15, 2969-2983, https://doi.org/10.5194/acp-15-2969-2015.
* * *

---

## Author Comment (AC1) · 18 Jan 2018

**Comments and Responses**

Referee #1

**Comments:** In the manuscript, the study reported the first observations of PM2.5_$\Delta^{17}O(SO_4^{2-})$ during haze events from October 2014 to January 2015 in Beijing, and use them to quantify the relative importance of different sulfate formation pathways, which is quite interesting and significant. But there are some mistakes and problem:

**Responses:** Thanks for your comments. We reply to your comments one by one as follows:

1.  **Q:** There are some typesetting in the manuscript: such as: (1) line 1: should be sulfate production; (2) line 25: formationwith should be formation with;(3) line 30: should be $NO_2$ in; (4) line 84: should be round 20m; (5) line 84:should be round 60m; line 243: should be Figure 1a shows;

   **A:** Thanks for noticing these mistakes. Some typesetting problems occurred when we used another computer to edit and upload the document in final. We corrected all the typesetting problems you mentioned. The corrected typesetting is respectively in line 1 for "sulfate production", in line 30 for "$NO_2$ in", in line 87 for "around 20 m", in line 87 for "around 60 km" and in line 248 for "Figure 1a shows".

2.  **Q:** Sampling site locate at a country-site (is not downtown), round 60 km northeast of downtown, and so that it is not perfect to know isotopic constraints on heterogeneous sulfate production in Beijing haze, the reason is that downtown has more information about the automobile exhaust emission, enterprise emission, and resident emission and so on. Furthermore, the sampling site is close to Yanxi lake, so that it has been probably affected by cloud liquid water content;

   **A:** Thanks for your comments. For your concern about the sampling site, we do agree that it would be better if the sampling site was in downtown area. However, we think our sampling site is a representative site of Beijing haze for the following reasons:

(1)  High concentrations of both secondary and primary air pollutants were observed at our sampling site. Figure 1 in the main text shows that 12h-averaged $SO_4^{2-}$ concentrations at our sampling site can be up to 56.4 μg m$^{-3}$. The 1h-averaged concentrations of primary air pollutants such as CO at our sampling site can be as high as 4025 ppb (Fig. C1), and shows similar trends and range with observations at the nearest station (Huairou station) set for estimating urban environment by Beijing Municipal Environmental Monitoring Center (BJMEMC, http://zx.bjmemc.com.cn/getAqiList.shtml?timestamp=1513326206397). For other air pollutants that were not observed at our sampling site but observed at BJMEMC stations (e.g., PM$_{2.5}$, $SO_2$ and $NO_2$), the comparisons between observations at Huairou station and a downtown site (Tiantan station, Fig. C2) show that the variations and range of PM$_{2.5}$ and $SO_2$ are very similar in these two sites. We also note that the peak concentrations of CO and $NO_2$ at or around our sampling site are generally lower than those observed at Tiantan station, so we do agree as you comment "downtown has more information about the automobile exhaust emission, enterprise emission, and resident emission and so on". However, our sampling site is suggested to be a suburban site based on the above comparisons and other studies (Wang et al., 2016; Wang et al., 2017; Li

et al., 2017; Tong et al., 2015).

(2) Our sampling site is usually downwind of downtown Beijing during polluted days (PD, $PM_{2.5} > 75$ μg m$^{-3}$). Backward trajectory analysis shows that air masses from downtown Beijing can reach our sampling site within a day during PD (Fig. C3), which suggests our sampling site can reflect the main signal of downtown Beijing, especially considering the lifetime of sulfate which is 4–5 days (Alexander et al., 2012).

(3) The total area of Beijing is around 16411 km$^2$ while its urban area is only about 1401 km$^2$. As haze in Beijing and North China Plain is a regional phenomenon (Zheng et al., 2015), it's truly important to sample at downtown area, however, it may be also necessary to sample at suburban area to have a more comprehensive understanding of Beijing haze.

(4) Our sampling site at UCAS (University of the Chinese Academy of Sciences) is a supersite set by HOPE-J$^3$A (Haze Observation Project Especially for Jing-Jin-Ji Area). Other observations at this site have been used to discuss scientific problems in Beijing haze in previous studies, e.g., (Zhang et al., 2017; Xu et al., 2016; Chen et al., 2015; Tong et al., 2015). Especially, in the work of Tong et al. (2015), they found that the heterogeneous reaction efficiency from $NO_2$ to HONO is higher in suburban areas than urban areas. As our sampling site is the same as the suburban site in the work of Tong et al. (2015), we think our sampling site should be suitable, even though it may be not perfect, to discuss heterogeneous sulfate production in Beijing haze.

(5) It's true that our sampling site is close to Yanqi lake (Fig. C2). The linear distance from our sample site to the nearest edge of Yanqi lake is around 1 km and is around 3 km to the farthest edge of Yanqi lake. We note that it's difficult to give an accurate estimate of the influence of Yanqi lake on cloud liquid water in the present study as the formation of cloud is quite complicated. However, as the area of Yanqi lake is about 2.3 km$^2$, the area fraction of Yanqi lake to the nearest 3 km circular area from our sample site is about 8 % (=2.3/(3.14$\times 3^2$)), which is small. So that we expect its influence on cloud liquid water is probably small, too, especially in winter when the lake is frozen.

[Figure]

**Figure C1.** Comparisons of air pollutants observed at our sampling site with that observed at Huairou station and Tiantan station. The Huairou station is the nearest station (to ours) set for estimating urban environment by Beijing Municipal Environmental Monitoring Center (BJMEMC), while the Tiantan station is located in the center of downtown Beijing (please refer to Fig. C2 as below). The missing concentrations of CO at our sampling site are due to that our CO analyzer (EC9830B, Ecotech Inc., Australia) were taken way to be calibrated. Hourly concentrations of other air pollutants (e.g., $PM_{2.5}$, $SO_2$ and $NO_2$) were not observed at our sampling site but observed at BJMEMC stations.

[Figure]

**Figure C2.** Map of our sampling site. The blue area on the base map represents rivers and lakes with a map scale of 1:1250000.

[Figure]

**Figure C3.** 1-day backward trajectories reaching our sampling site at each hour during our sampling time when $PM_{2.5} \geq 75$ $\mu g\ m^{-3}$.

3. **Q:** Method: As we know, these data are likely contaminated to various degrees by occluded nitrate isotope signal in the samples, so that we had better remove nitrate during the preparing the $Ag_2SO_4$, but the manuscript has not removed the nitrate, which will effect on the sulfur isotope composition results;

  **A:** Thanks for your comment. Actually, we removed $NO_3^-$ before $SO_4^{2-}$ being converted to $Ag_2SO_4$. The preparation and measurements of $\Delta^{17}O(SO_4^{2-})$ in the University of Washington have been well documented in previous studies (Geng et al., 2013; Chen et al., 2016; Alexander et al., 2012), thus we didn't describe the experimental procedures in detail in the former manuscript. Now, the description "$SO_4^{2-}$ was separated from other anions (e.g., $NO_3^-$) by ion chromatography" is presented in lines 111–112.

4. **Q:** The manuscript should introduce name of laboratory, precision of the machine.

  **A:** Thanks for your suggestions. The name of the laboratory where ions were measured now is added in lines 98–99 reading "The measurements of ions were conducted in Anhui Province Key Laboratory of Polar Environment and Global Change in University of Science and Technology of China". The description about precisions of ion measurements is added in lines 104–105 reading "Typical analytical precision by our instrument is better than 10 % RSD (relative standard deviation) for all ions (Chen et al., 2016)". The name of the laboratory where our $\Delta^{17}O(SO_4^{2-})$ was measured now is added in lines 105–107 reading "The preparation and measurements of $\Delta^{17}O(SO_4^{2-})$ were conducted in Isolab (https://isolab.ess.washington.edu/isolab/) at the University of Washington, USA". The precision is dependent on the instrument itself and the method. The description about precision of our $\Delta^{17}O(SO_4^{2-})$ measurements (which includes the machine MAT253) now is added in lines 116–118 reading "The precision of $\Delta^{17}O$ measurements in our method is ±0.3 ‰ based on replicate analysis of standards, which is consistent with previous studies (Alexander et al., 2005; Sofen et al., 2014; Chen et al., 2016)".

**References**

Alexander, B., Park, R. J., Jacob, D. J., Li, Q., Yantosca, R. M., Savarino, J., Lee, C., and Thiemens, M.: Sulfate formation in sea‐salt aerosols: Constraints from oxygen isotopes, J. Geophys. Res., 110, D10307, 2005.

Alexander, B., Allman, D., Amos, H., Fairlie, T., Dachs, J., Hegg, D. A., and Sletten, R. S.: Isotopic constraints on the formation pathways of sulfate aerosol in the marine boundary layer of the subtropical northeast Atlantic Ocean, J. Geophys. Res., 117, D06304, 2012.

Chen, Q., Geng, L., Schmidt, J. A., Xie, Z., Kang, H., Dachs, J., Cole-Dai, J., Schauer, A. J., Camp, M. G., and Alexander, B.: Isotopic constraints on the role of hypohalous acids in sulfate aerosol formation in the remote marine boundary layer, Atmos. Chem. Phys., 16, 11433-11450, 2016.

Chen, Z., Zhang, J., Zhang, T., Liu, W., and Liu, J.: Haze observations by simultaneous lidar and WPS in Beijing before and during APEC, 2014, Sci. China Chem., 58, 1385-1392, 2015.

Geng, L., Schauer, A. J., Kunasek, S. A., Sofen, E. D., Erbland, J., Savarino, J., Allman, D. J., Sletten, R. S., and Alexander, B.: Analysis of oxygen‑17 excess of nitrate and sulfate at sub‑micromole levels using the pyrolysis method, Rapid Commun. Mass Spectrom., 27, 2411-2419, 2013.

Li, K., Li, J., Wang, W., Tong, S., Liggio, J., and Ge, M.: Evaluating the effectiveness of joint emission control policies on the reduction of ambient VOCs: Implications from observation during the 2014 APEC summit in suburban Beijing, Atmos. Environ., 2017.

Sofen, E., Alexander, B., Steig, E., Thiemens, M., Kunasek, S., Amos, H., Schauer, A., Hastings, M., Bautista, J., and Jackson, T.: WAIS Divide ice core suggests sustained changes in the atmospheric formation pathways of sulfate and nitrate since the 19th century in the extratropical Southern Hemisphere, Atmos. Chem. Phys., 14, 5749-5769, 2014.

Tong, S., Hou, S., Zhang, Y., Chu, B., Liu, Y., He, H., Zhao, P., and Ge, M.: Comparisons of measured nitrous acid (HONO) concentrations in a pollution period at urban and suburban Beijing, in autumn of 2014, Sci. China Chem., 58, 1393-1402, 2015.

Wang, G., Cheng, S., Wei, W., Zhou, Y., Yao, S., and Zhang, H.: Characteristics and source apportionment of VOCs in the suburban area of Beijing, China, Atmos. Pollut. Res., 7, 711-724, 2016.

Wang, Y., de Foy, B., Schauer, J. J., Olson, M. R., Zhang, Y., Li, Z., and Zhang, Y.: Impacts of regional transport on black carbon in Huairou, Beijing, China, Environ. Pollut., 221, 75-84, 2017.

Xu, X., Zhao, W., Zhang, Q., Wang, S., Fang, B., Chen, W., Venables, D. S., Wang, X., Pu, W., and Wang, X.: Optical properties of atmospheric fine particles near Beijing during the HOPE-J 3 A campaign, Atmos. Chem. Phys., 16, 6421-6439, 2016.

Zhang, J., Chen, Z., Lu, Y., Gui, H., Liu, J., Liu, W., Wang, J., Yu, T., Cheng, Y., and Chen, Y.: Characteristics of aerosol size distribution and vertical backscattering coefficient profile during 2014 APEC in Beijing, Atmos. Environ., 148, 30-41, 2017.

Zheng, B., Zhang, Q., Zhang, Y., He, K., Wang, K., Zheng, G., Duan, F., Ma, Y., and Kimoto, T.: Heterogeneous chemistry: a mechanism missing in current models to explain secondary inorganic aerosol formation during the January 2013 haze episode in North China, Atmos. Chem. Phys., 15, 2031-2049, 2015.

---

## Author Comment (AC2) · 18 Jan 2018

**Comments and Responses**

Referee #2

**Comments:** This manuscript presents a new theoretical framework for quantifying heterogeneous sulfate production pathways in Beijing haze using field-based measurements of mass independent compositions of oxygen isotopes ($\Delta^{17}O$) in sulfates. The dataset is the first measurement of $\Delta^{17}O(SO_4^{2-})$ in fine particles in the megacity Beijing. In addition, a combination of metastable and stable states was proposed to calculate the aerosol acidity, which plays a predominant role in the relative contributions of $O_2$ and $NO_2$ oxidation pathways. The sulfate formation in Beijing haze is a subject of intense scrutiny in recent years in atmospheric chemistry community, and the use of triple oxygen isotopic analysis for such quantification is a large step forward in this field. The isotopic measurements made in UW are of high quality as usual. Some uncertainties exist in calculation and deserve further assessment, but given that this is a first investigation and the manuscript is thorough and interesting, I strongly recommend publication in ACP after considering the following comments and suggestions.

**Response:** Thanks for your comments. We reply to your comments one by one as follows:

Major comments:

1.    **Q:** In this manuscript, there is a big assumption that the $\Delta^{17}O$ of sulfates produced by the $NO_2$ oxidation is zero. I think the authors are probably right, but there remain uncertainties because the $NO_2$ oxidation mechanism has not yet been defined. In the introduction, the authors cited the work of Shen and Rochelle (1998) who proposed a radical chain reaction. In this case, I agree that the sulfate product is normal. In the other work cited by the author (He et al., 2014), it was proposed that oxygen is a key oxidant and oxygen atom transfer from $O_2$ to $SO_4^{2-}$ via $NO_2$. The conclusion made by He et al. (2014) is based on a set of laboratory experiments, in which sulfates would not produce without $O_2$. However, in their experiments, the role of $O_3$ was not examined. In the ambient atmosphere (especially in urban areas), the reaction $NO+O_3 \rightarrow NO_2+O_2$ cannot be ignored. Although it was argued that $O_3$ was not important in Beijing haze because of its low mixing ratio, is it possible that the low $O_3$ mixing ratio is a result of enhanced $NO+O_3 \rightarrow NO_2+O_2$ reaction (aka "titration effect")? As shown in many studies (e.g., Xu et al., 2011; Zhang et al., 2015), ozone mixing ratios in the urban region are lower than the surrounding rural region (probably in part due to large local emissions of fresh NO in the urban region). In this case, the oxygen anomaly in ozone molecules would lead to positive $\Delta^{17}O$ values in sulfates produced via the $NO_2$ oxidation. Because of the enriched $^{17}O$ in ozone, a small fraction can lead to a non-zero $\Delta^{17}O$ in the sulfate product. In addition, $NO+HO_2 \rightarrow OH+NO_2$ is also a possible pathway to transfer anomalous oxygen atoms (with the assumption that $\Delta^{17}O$ in $HO_2$ is non-zero based on $\Delta^{17}O$ values in $H_2O_2$). Can the authors provide a quantitative estimation on the possible contribution of $O_3/HO_2$ to the $NO_2$ oxidation? Because the validity of this assumption can significantly alter the conclusion of this paper and there is no laboratory experiment to support this assumption at present, I think such discussion would make the authors' case stronger.

**A:** Thanks for your comments. In the work of He et al. (2014), they stated in their paper that "As shown in Fig. 2, sulfate can be formed on the CaO surface only in the presence of $O_2$. Similar phenomena were also found on $Al_2O_3$, ZnO, and $TiO_2$ surfaces (see Supplementary Information). Therefore, $O_2$ was the key oxidant in the process of $SO_2$ oxidation". This statement should be reliable as it is what's shown in Fig. 2 of their work (this figure is presented below for convenience). But I think the mechanism of $SO_2$ oxidation proposed by them is contradictory to this figure. In their proposal, the mechanism of $SO_2$ oxidation has two steps, the first one is ($SO_2+2NO_2+M\rightarrow M$–$SO_4+2NO$) and the second one is ($2NO+O_2+M\rightarrow 2NO_2$). It should be noted that $SO_4^{2-}$ is thought to form in the first step of their proposed mechanism, with two oxygen-atom directly from $NO_2$ without $O_2$. If this mechanism is correct, we expect $SO_4^{2-}$ being seen when $SO_2+NO_2$ is exposed to the surface of CaO. However, in their laboratory experiments, when they continuously exposed $SO_2+NO_2$ to the surface of CaO, $SO_4^{2-}$ was not observed at all (see black solid squares in Fig. 2 of their work). Therefore, I think the two-steps oxidation mechanism that they proposed to explain the experimental results in their study is problematic. One more piece of evidence supports our speculation and is shown in Fig. 1B of their work (this figure is also presented below for convenience). Sulfite but not sulfate is observed when they exposed $SO_2$ to the surface of CaO, which means one oxygen-atom from $H_2O$ is transferred to sulfite. $SO_4^{2-}$ was not observed when it was exposed only to $NO_2$ but was observed when continually exposed to $NO_2+O_2$, again suggesting that the oxygen-atom cannot be directly transferred from $NO_2$ but from $O_2$. In summary, in the oxidation of $SO_2$ to $SO_4^{2-}$, one oxygen-atom is transferred from $H_2O$ to form S(IV) (= $SO_2 \cdot H_2O+HSO_3^-+SO_3^{2-}$), the other is from $O_2$ but not via $NO_2$ based on the experimental results of He et al. (2014). As for the specific mechanism for $SO_2+NO_2+O_2$ in the experiments of He et al. (2014), it may be similar (but different) to the proposed mechanism by Clifton et al. (1988). In the work of Clifton et al. (1988), the oxidation mechanism of S(IV) by $NO_2$ in bulk solution was proposed as follows:

$2NO_2+SO_3^{2-}\rightarrow(O_2N$–$SO_3$–$NO_2)^{2-}$

$(O_2N$–$SO_3$–$NO_2)^{2-}+OH^-\rightarrow(HO$–$SO_3$–$(NO_2)_2)^{3-}$

$(HO$–$SO_3$–$(NO_2)_2)^{3-}\rightarrow 2NO_2^-+SO_4^{2-}+H^+$

Similarly, we propose the experimental results in the work of He et al. (2014) can be explained as follows: when $SO_2$ is exposed the surface of CaO, sulfite formed (Fig. 1B of their work). In the presence of $NO_2$, sulfite may react with $NO_2$ to form an addition complex (e.g., $(O_2N$–$SO_3$–$NO_2)^{2-}$), so $SO_4^{2-}$ is not observed (black solid squares in Fig. 2 of their work). In the presence of $O_2$, the formed addition complex may react with $O_2$ (or oxygen radicals induced by $O_2$, e.g., $O_2^-$, $O^-$) to form $SO_4^{2-}$ (red hollow squares in Fig. 2 of their work). So even though we agree with the experimental results of He et al. (2014) that "$O_2$ was the key oxidant in the process of $SO_2$ oxidation", we think the oxygen-atom transfer from $O_2$ may be via an addition complex but not via $NO_2$. It's worth noting that the proposed oxidation mechanism for S(IV)+$NO_2$, no matter via a radical chain mechanism (Shen and Rochelle, 1998) or via oxygen-atom transfer from $OH^-$ (Clifton et al., 1988) or via oxygen-atom transfer from $O_2$ in our proposal based on experimental results of He et al. (2014), all result in $\Delta^{17}O(SO_4^{2-})$ =

0 ‰. So based on information available in literature, we think it's appropriate to assume S(IV)+NO$_2$ leads to $\Delta^{17}O(SO_4^{2-})$ = 0 ‰. Therefore we describe this as "Sulfate produced by NO$_2$ oxidation is suggested to occur either via a radical chain mechanism (Shen and Rochelle, 1998), via oxygen-atom transfer from OH⁻ (Clifton et al., 1988), or from O$_2$ based on experimental results of He et al. (2014), resulting in $\Delta^{17}O(SO_4^{2-})$ = 0 ‰" in lines 74–76 of the present manuscript.

Since it is not oxygen in NO$_2$ that is directly transferred to sulfate, the role of O$_3$/HO$_2$ in NO oxidation is irrelevant in this case.

[Figure]

**Figure 1 | Surface coverage of (A) sulfate and (B) sulfite species on mineral oxides after heterogeneous reaction of SO$_2$ or SO$_2$+NO$_2$ for 2 h.** Reaction conditions: carrier gases: N$_2$(80%) + O$_2$(20%) with total flow of 100 mL/min; concentrations of SO$_2$ and NO$_2$: both 200 ppm; T = 303 K. A synergistic effect between SO$_2$ and NO$_2$ was also observed when reactant concentrations were at ppb level with long reaction time (see Supplementary Information).

Figure 1 of the work of He et al. (2014).

[Figure]

**Figure 2 | Comparison of integral infrared peak area of sulfate when NO$_2$+SO$_2$ was exposed to the surface of CaO with pure N$_2$ (black solid squares) and synthetic air (80%N$_2$+20%O$_2$) (red hollow squares) as carrier gas.** Reaction conditions: total flow = 100 mL/min; concentrations of SO$_2$ and NO$_2$ both 200 ppm; T = 303 K.

Figure 2 of the work of He et al. (2014).

2.    **Q:** The calculation of pH is a little bit outside of my area of expertise, but I think the authors did a good job of discussing uncertainties and caveats, and the calculation of metastable/stable states proposed by the authors seems scientifically sound. My only concern is the discussion of NO$_2$/O$_2$ oxidation pathways (Lines 341-369). I understand the authors want to convince readers that the NO$_2$ oxidation could be potentially important. However, as noted by the authors,

the estimated production rate of $O_2$ oxidation is _4 orders of magnitude greater the $P_{het}$ (lines 358-369). Therefore, a very small fraction of aerosols with pH<3 seems enough to explain the heterogeneous production rate via $O_2$ oxidation. In this case, why do we need the $NO_2$ oxidation? I would like to see discussion why $O_2$ oxidation cannot explain the heterogeneous formation here and why $NO_2$ oxidation is required. I think discussing the $O_2$ oxidation first and then the $NO_2$ oxidation would make this part easier to read and follow.

**A:** Thanks for your affirmations and comments. One of our points is that $NO_2$ and $O_2$ oxidation pathways may co-exist in ambient atmosphere due to the heterogeneity of aerosol state and pH. Therefore, we discussed that at different conditions the importance of $NO_2$ or $O_2$ oxidation pathway in heterogeneous sulfate production varies. In our discussion, we attempted to pay equal attention to these two pathways. As you know, however, no other experimental results about $SO_2$ oxidation by $O_2$ on acidic microdroplets have been published yet beyond the pioneering work of Hung and Hoffmann (2015), so we have little information about this pathway. Hung and Hoffmann (2015) reported the maximum of the reaction rate of $O_2$ oxidation on acidic microdroplets at pH $\leq$ 3 and suggested that it decreased with increasing pH when pH > 3 without reporting its specific value. However, the aerosol pH calculated by ISORROPIA II is far larger than 3 (7.6±0.1 for stable state assumption and 4.7±1.1 for metastable state assumption), so we cannot figure out its production rate but calculate its maximum value by using reaction rate at pH $\leq$ 3, and we use this maximum to see if it can meet our calculated $P_{het}$. The estimated maximum of $P_{het, S(IV)+O2}$ is four order of magnitude larger than $P_{het}$, which is too high. But, on the other hand, we note that in the work of He et al. (2014), $SO_4^{2-}$ was not seen when $SO_2+O_2$ was exposed on most mineral oxides (Fig. 1A of their work) and in the work of Wang et al. (2016), they found $SO_2$ oxidation by $O_2$ was negligible in ammonium solution. These two work along with the work of Hung and Hoffmann (2015) directly suggest $O_2$ oxidation pathway may be negligible at higher pH conditions. Since we cannot quantify the fraction of aerosols with pH $\leq$ 3 (Kelvin effect) and even cannot verify their existence in the ambient atmosphere during Beijing haze, we realize that the uncertainty of estimating the production rate of $O_2$ oxidation pathway is far larger than $NO_2$ oxidation pathway. Based on this situation, we first discussed the $NO_2$ oxidation pathway and when $NO_2$ oxidation was not high enough to meet $P_{het}$, we further examined $O_2$ oxidation on acidic microdroplets. To better remind readers of the uncertainty of estimating $O_2$ oxidation pathway in the present manuscript, we described "This value should be an overestimate due to our calculated bulk aerosol pH predicted in metastable state being (4.4±0.6) during PD and the experimental results of He et al. (2014) and Wang et al. (2016) suggests $O_2$ oxidation pathway is negligible at higher pH conditions (e.g., on CaO and in $NH_4^+$ solution)" in lines 369–372. In summary, we think $NO_2/O_2$ oxidation pathways should be paid equal attention to. The order of discussing $NO_2$ oxidation first and then $O_2$ oxidation in our present scenario is based on the consideration that there exists larger uncertainty of estimating production rate of $O_2$ oxidation pathway due to less information about this pathway. So more studies about $SO_2$ oxidation by $O_2$ on acidic microdroplets proposed by Hung and Hoffmann (2015) need to be done in the future.

Specific comments:

1.  **Q:** Line 24: Please give a quantitative context here (48+-5%?). The manuscript is focused on the heterogeneous sulfate production and therefore it is important to let readers know its overall contribution.

    **A:** Thanks for this suggestion. This information has been given in lines 24–26 now, it reads "However, heterogeneous sulfate production ($P_{het}$) on aerosols was estimated to dominate sulfate formation during PD of other cases, with a fractional contribution of $(48\pm5)$ %".

2.  **Q:** The introduction could be better constructed. As noted by the authors, the relative importance of $O_2$ and $NO_2$ oxidation pathways is highly depending on pH and is difficult to constrain (lines 60-62). In lines 76-77, the authors state that the relative importance of different sulfate formation pathways is quantified in this study. So when I read this part, I thought the authors successfully solved this problem. However, this is not the case. I think the major advantage of $\Delta^{17}O$ in this study is to constrain the $O_3$ oxidation pathway in heterogeneous sulfate formation, which is of large uncertainties in previous studies. This should be highlighted. In the end of introduction, it's better to explicitly state something like "the contributions of $O_3$ and $H_2O_2$ oxidation in heterogeneous sulfate formation are quantified, and the roles of $NO_2/O_2$ oxidation are discussed." To prevent any overstatement.

    **A:** Thanks for your suggestions. We added "Laboratory work has suggested $SO_2$ oxidation by $O_3$ on mineral dust is a significant pathway for sulfate production (Li et al., 2006), but its role in Beijing haze has not been determined." in lines 57–58 to pave the way for the following discussion. Based on your suggestions, we also state that "In this work, first observations of $PM_{2.5}$ $\Delta^{17}O(SO_4^{2-})$ during haze events from October 2014 to January 2015 in Beijing are reported, contributions of $O_3$ and $H_2O_2$ oxidation in heterogeneous sulfate formation are quantified, and the roles of $NO_2$ and $O_2$ oxidation are explored." in lines 78–80 to prevent any overstatement.

3.  **Q:** Lines 255-256: A recent study reported one-year $\Delta^{17}O$ measurements in sulfates collected from a background mountain site in East China (Lin et al., 2017). This work is closely connected to the subject of the manuscript and should be cited.

    **A:** Thanks for your recommendation. This important work has been cited in line 260 now.

4.  **Q:** Lines 259-260 and Fig. 2: This part is not clear to me. Do the authors mean that the $\Delta^{17}O$ is directly linked to the $O_3/H_2O_2$ concentrations? If this is the case, scatter plots (with correlation coefficients) or time series may be clearer. I am also confused why cases I and II were grouped together. In the rest part of this manuscript (e.g., abstract and Figure 5), case II seems significantly different from other cases. And where is case V? Please clarify. In addition, it is better to use "calculated $H_2O_2$" as the y-axis title of Fig. 2c.

    **A:** Thanks for your comments. The objective of this part is to see whether the observed results "the NPD to PD difference of $\Delta^{17}O_{obs}$ can be case-dependent." is roughly linked to some easily observed or estimated parameters (e.g., observed $O_3$ and calculated $H_2O_2$). We agree that "scatter plots (with correlation coefficients) or time series may be clearer",

so we grouped the time series of observed $O_3$ and calculated $H_2O_2$ in Fig. 1c and removed the former Fig. 2. We also show the scatter plots with correlation coefficients below (Fig. C1) for your review. For your concern about why we group Cases I and II together and Cases III to V together in the former Fig. 2, we do so based on two reasons. The first reason is the phenomenon that $\Delta^{17}O_{obs}$ increased from NPD to PD during Case I and II while the opposite trend was observed for Case III to V (Fig. 1b). The second reason is that Case I and II is in autumn without centralized heating while Case III to V is during the heating season in Beijing. In addition, we did show the data of Case V in the former Fig. 2, I think you may not notice the x-axis title in the right part of former Fig. 2, which reads "Case III to V", or I may misunderstand your comment. As you suggested, we used "calculated $H_2O_2$" as the y-axis title of Fig. 1c in the present manuscript.

[Figure]

**Figure C1.** The relationship between $O_3$ and $\Delta^{17}O_{obs}$ **(a)** and relationship between calculated $H_2O_2$ and $\Delta^{17}O_{obs}$ **(b)**. The red lines are linear least-squares fitting lines.

5.  **Q:** Line 263: What is "the range of any single reaction pathway"? It is not clear to me. I don't think this statement is exactly correct. For example, a sample with $\Delta^{17}O$ ranging from 0.6 to 1 per mil could be 100% produced from the $H_2O_2$ oxidation because $\Delta^{17}O$ values in $H_2O_2$ are in the range of 1.2-2 per mil. The observed $\Delta^{17}O$ value is not a supportive evidence for this statement.

    **A:** Thanks for noticing this. We realize this statement is not exactly correct and have removed it now.

6.  **Q:** Lines 274-276: Why did the author look at the $PM_{2.5}$ instead of sulfate concentration? I think a good correlation between $P_{het}$ and sulfate concentrations would be more convincing. From Figure 5, it seems that the variation of $SO_4^{2-}$ is more correlated to $P_{cloud}$ than $P_{het}$. The similarity of $\Delta^{17}O_{cloud}$ and $\Delta^{17}O_{obs}$ in Figure 6 also likely indicates that the contribution of $P_{cloud}$ is more dominant than $P_{het}$. If this is the case, the role of $P_{het}$ may be overstated. I would like to see a table showing the percentages of $P_{het}$, $P_{cloud}$ and $P_{OH}$ in each case.

    **A:** Thanks for your comments. We look at the relationship between $PM_{2.5}$ concentration and $P_{het}$ instead of sulfate with $P_{het}$ in former Fig. 4b of our former manuscript due to that we want to examine $P_{het}$ variations during the evolution of haze pollution, where haze events are defined based on $PM_{2.5}$ concentrations. Based on your suggestions, we removed the relationship between $PM_{2.5}$ and $P_{het}$ and show the time series of $P_{het}$ along with $SO_4^{2-}$ concentrations in present Fig. 3b (also

shown as below). It's clear in Fig. 3b that $P_{het}$ shows very similar trends with $SO_4^{2-}$ concentrations except for Case II, in which sulfate production was found to predominantly occur by in-cloud reactions. Generally, the variation of $SO_4^{2-}$ along with $P_{cloud}$ is not as good as that with $P_{het}$ (Fig. C2) in our estimate. The estimated fractional contribution of different pathways is shown in Table C1 (also shown as Table S7 in SI). The $f_{het}$ and $f_{cloud}$ during polluted days in Case II is respectively 23 % and 68 % while $f_{het}$ and $f_{cloud}$ is respectively (48±5) % and (38±7) % in Case I and III to V, so I think it's not inappropriate to state that heterogeneous reactions were found to dominate sulfate formation in four out of the total five cases. For the similarity between $\Delta^{17}O_{cloud}$ and $\Delta^{17}O_{obs}$ (r = 0.63, p < 0.01), I think it mainly suggests that our estimate of in-cloud reactions should be reliable.

[Figure]

**Figure 3 in the main text.** The relationship between RH and SOR **(a)** and time series of overall heterogeneous sulfate production ($P_{het}$) along with $SO_4^{2-}$ concentrations **(b)**. The black line in (a) is linear least-squares fitting line.

[Figure]

**Figure C2.** Scatter plots of $P_{het}$ **(a)** and $P_{cloud}$ **(b)** with $SO_4^{2-}$. The light gray dots are during Case II.

**Table C1.** Estimated fractional contribution of different sulfate production pathways during Beijing haze.

| Case | $f_p$ (%)[a] | $f_{het}$ (%) | $f_{cloud}$ (%) | $f_{SO2+OH}$ (%) |
|------|------|------|------|------|
| I | 9 | 54 | 29 | 8 |
| II | 6 | 23 | 68 | 3 |

| | | | | |
|---|---|---|---|---|
| III | 11 | 42 | 46 | 1 |
| IV | 15 | 47 | 37 | 1 |
| V | 9 | 49 | 41 | 1 |

[a] $f_p$, $f_{het}$, $f_{cloud}$, and $f_{SO2+OH}$ respectively represents fractional contribution from primary sulfate, heterogeneous reactions, in-cloud reactions and gas-phase pathway.

7. **Q:** Lines 284-286: In Figure 5, the peak of $P_{cloud}$ is at 10/24, not exactly matching the $SO_4^{2-}$ peak at 10/25. Could the authors discuss about this difference? Is it because of a stagnant meteorological condition? Or is it possible that the $P_{cloud}$ was underestimated?

**A:** Thanks for your comment. Previous studies have suggested the stagnant meteorological condition is a key factor in the formation of haze in winter Beijing (Zheng et al., 2015), however, its role has not been quantified yet. For the reasons for the phenomenon that the peak of $P_{cloud}$ is not exactly matching $SO_4^{2-}$ peak at 10/25, in addition to the stagnant meteorological condition as you comment, another reason may be that the large mass of sulfate produced in cloud needs to accumulate to lead to the peak of surface $SO_4^{2-}$. It could be possible that in-cloud reactions were underestimated; however, previous work by Wang et al. (2014) using a global model showed that biases in meteorology, including biases in clouds, cannot explain the models underestimate of sulfate production rates during a Beijing haze event.

8. **Q:** Line 371-372: Please give a quantitative context as suggested before.

**A:** Thanks for this suggestion. A quantitative context has been added, it reads "Our study suggests that both in-cloud reactions and heterogeneous reactions can dominate sulfate formation during Beijing haze, with the fractional contribution of $f_{cloud} = 68\%$ in Case II and $f_{het} = (48\pm5)$ % in Case I and III–V" in lines 376–377.

Typos:

9. **Q:** Line 84: "around"

**A:** Thanks for noticing this. The word "round" has been changed into "around" and the missing space has been added in line 87.

10. **Q:** Line 149: "sulfate formation"

**A:** Thanks for noticing this. The missing space between "sulfate" and "formation" has been added in line 154.

11. **Q:** Line 282: "cases"

**A:** Thanks for noticing this. The word "Cases" has been changed into "cases" in line 286.

12. **Q:** There are many spaces missing in the manuscript. I am not going to go through all of them.

**A:** Thanks for noticing this. We are very sorry that many spaces missed when we use another computer to edit and upload the document in final. We added all the missing space in the present manuscript.

**References**

Clifton, C. L., Altstein, N., and Huie, R. E.: Rate constant for the reaction of nitrogen dioxide with sulfur (IV) over the pH range 5.3-13, Environ. Sci. Technol., 22, 586-589, 1988.

He, H., Wang, Y., Ma, Q., Ma, J., Chu, B., Ji, D., Tang, G., Liu, C., Zhang, H., and Hao, J.: Mineral dust and NOx promote the conversion of SO2 to sulfate in heavy pollution days, Sci. Rep., 4, 4172, 2014.

Hung, H.-M., and Hoffmann, M. R.: Oxidation of gas-Phase SO2 on the surfaces of acidic microdroplets: Implications for sulfate and sulfate radical anion formation in the atmospheric liquid phase, Environ. Sci. Technol., 49, 13768-13776, 2015.

Li, L., Chen, Z., Zhang, Y., Zhu, T., Li, J., and Ding, J.: Kinetics and mechanism of heterogeneous oxidation of sulfur dioxide by ozone on surface of calcium carbonate, Atmos. Chem. Phys., 6, 2453-2464, 2006.

Shen, C. H., and Rochelle, G. T.: Nitrogen dioxide absorption and sulfite oxidation in aqueous sulfite, Environ. Sci. Technol., 32, 1994-2003, 1998.

Wang, G., Zhang, R., Gomez, M. E., Yang, L., Zamora, M. L., Hu, M., Lin, Y., Peng, J., Guo, S., and Meng, J.: Persistent sulfate formation from London Fog to Chinese haze, P. Natl. Acad. Sci. USA, 113, 13630-13635, 2016.

Wang, Y., Zhang, Q., Jiang, J., Zhou, W., Wang, B., He, K., Duan, F., Zhang, Q., Philip, S., and Xie, Y.: Enhanced sulfate formation during China's severe winter haze episode in January 2013 missing from current models, J. Geophys. Res., 119, 10425-10440, 2014.

Zheng, B., Zhang, Q., Zhang, Y., He, K., Wang, K., Zheng, G., Duan, F., Ma, Y., and Kimoto, T.: Heterogeneous chemistry: a mechanism missing in current models to explain secondary inorganic aerosol formation during the January 2013 haze episode in North China, Atmos. Chem. Phys., 15, 2031-2049, 2015.

---

## Editor Decision (ED1)

Dear authors,

Many thanks for your revised submission. Please take note of the reviewers' comments and my comments below when your prepare a revised manuscript.

There are various problems with the dimensions of quantities in equations 1, 4, 5 and 8, and in case of equations 5 and 8, this may result in large changes to some of your results.

Section 2.7 does not state how the various fractions $f$ have been calculated. Please give explicit equations that clarify this, including what the input terms (reaction rates, $\Delta(^{17}O)$ values) are and how these input terms have been calculated themselves.

The reviewers have also raised a number of points that need to be addressed. In particular, points 1 and 2 of reviewer 2 (source of the air; role of O3 as oxidant) need some careful discussion. Table S7 should be moved to the main text.

There are still a few problems with missing units in equations – please refer to the first chapter of the IUPAC Green Book (https://www.iupac.org/fileadmin/user_upload/publications/e-resources/ONLINE-IUPAC-GB3-2ndPrinting-Online-Sep2012.pdf) or chapter 5 of the SI brochure (https://www.bipm.org/en/publications/si-brochure/) for examples of correct quantity notation.

The term "concentration" is not interchangeable with "mole fraction". Please use the term "mole fraction" where you refer to the latter (e.g. l. 197 and 198).

Data availability: Please include a table with the data from Figures 1, 4, 5 and 6 and the individual input values used for each sample in the ISORROPIA model.

Yours sincerely
Jan Kaiser

l. 67: A quantity symbol (e.g. $R$) must be used to define the isotope ratios and the index must follow immediately after the quantity symbol, e.g. $R_{sample}(^xO/^{16}O)$. "$X$" should be written in italics because it is a quantity symbol.

l. 92: Please replace "ppb" with the corresponding SI unit "nmol mol$^{-1}$", throughout the manuscript. Atmospheric Chemistry and Physics requires the use of SI units. Also, please write the equation in line with the rules of quantity algebra, i.e. $[H_2O_2] / (nmol\ mol^{-1}) = 0.1155\ e^{0.0846 T/^oC}$.

l. 125/Eq. 1: Please use quantity algebra for all equations, see IUPAC Green Book (https://www.iupac.org/fileadmin/user_upload/publications/e-resources/ONLINE-IUPAC-GB3-2ndPrinting-Online-Sep2012.pdf). Where quantities are given as explicit values, they must carry units (e.g. 96 g mol$^{-1}$, 3600 s h$^{-1}$).

Eq. 1 is dimensionally not correct; it has units of $\text{g m}^{-3}\text{ h}^{-1}\text{ atm}^{-1}$, but is supposed to have $\text{g m}^{-3}\text{ h}^{-1}$. Presumably the equation needs to include atmospheric pressure.

l. 127: The non-SI unit "atm" should be replaced with an SI-accepted unit, e.g. bar or Pa (or a derivative of them).

l. 129: The uptake coefficient has the unit "1"; it is not "unitless".

l. 132: Again, units are missing from this equation. Also, the quantity that "PM2.5" refers to must be identified, e.g. $\gamma(\text{PM2.5})$ or $\rho(\text{PM2.5})$, if it is a mass concentration. Both symbols are not ideal because they clash with the uptake coefficient and the bulk density. Perhaps the uptake coefficient should be given a different symbol than $\gamma$.

l.136: The quantity that PM2.5 refers to must be identified, e.g. $\gamma(\text{PM2.5})$. The extraneous factor $10^{-6}$ and the multiplication symbols ($\times$) should eliminated from the equation.

l. 137 & 205: Please choose a suitable single-letter symbol for relative humidity in these equations, e.g. $\Psi$.

l. 152: see l. 125: "$3600\text{ s h}^{-1}$","$96\text{ g mol}^{-1}$"; correct dimensions (presumably multiplication by atmospheric pressure).

l. 160 & 246: These equations is wrong in a bad way. The units on the right hand side are "$\text{g}^2\text{ m}^{-3}\text{ h}^{-1}$", but are supposed to be "$\text{g m}^{-3}\text{ h}^{-1}$". Again, it should be "$3600\text{ s h}^{-1}$" and "$96\text{ g mol}^{-1}$". Finally, the SI requires quantity symbols to consist of a single (Latin or Greek) letter, so LWC is not an acceptable symbol in an equation and should be replaced by a suitable one. (LWC as an abbreviation is fine, just not as a quantity symbol).
These errors suggest that $P_{\text{cloud}}$ values may be fundamentally wrong. Please discuss, using numerical examples, the impact of correcting the equation on your results.

l. 167 & l. 180: Include "‰" after 6.5 and 0.7

l. 196: This equation requires quantity symbols for the mole fractions and the unit "$\text{nmol mol}^{-1}$" needs to appear in the right place, e.g. "$y(\text{NH}_3) = 0.34y(\text{NO}_x) + 0.63\text{ nmol mol}^{-1}$".

l. 197: Replace "concentration" with "mole fraction" – also other occurrences of the word "concentration" in the text may need to be replaced with "mole fraction". Concentration implies an amount per volume.

l. 205: "MF" should be replaced with a suitable single-letter symbol, e.g. "$x(\text{metastable})$".

l. 211: The terms involving logarithms of concentrations and ion strengths in equations need to be divided by the standard concentration ($c^{\ominus} = 1\text{ mol dm}^{-3}$), to make them dimensionally correct. The units of $\beta^*$ need to be identified.

l. 258: Please use appropriate symbols, e. g $c(\text{SO}_4^{2-})$ for sulfate concentrations.

p. 11: Remove unnecessary brackets around pH expressions, e. g. 7.6±0.1. The brackets are only required where similar such expressions have units.

Figure 2: In the figure caption, please include an explicit link to the newly added equations in the main text that give explicit solutions for the fractions shown in this figure.

Table S1:
$k_{0low}$ = 3.3×10$^{-31}$ ($T$/300 K)$^{-4.3}$ cm$^6$ s$^{-1}$ [$T$ has units of K; molecule is not a unit]
$k_{0high}$ = 1.6×10$^{-12}$ cm$^3$ s$^{-1}$ [since ($T$/300 K)$^0$ = 1]

Table S4:
Did you only use these mean values in your thermodynamic calculations? Or did you use sample-specific input parameters?

---

## Author Response (AR2)

**Response to reviewers' and Editors' Comments**

**Reviewer 1 comments.**

I have the following suggestions that I urge the authors to consider in their final version.

**1. Q**: A major issue in this paper is that a regional reaction-transport model is not used. It has been demonstrated that in Beijing, under many meteorological conditions, a large quantity of the atmospheric pollutants can be transported in from the south. Thus, the in-cloud water and aerosol water parameters that were responsible for the secondary atmospheric sulfate formation in Beijing are in fact conditions a couple of days older and of different locations. This point is not considered in the text and should be added to the discussion and to be mentioned as a caveat in the abstract.

A: Thanks for the comment. It's true that the polluted air mass could have been processed under haze conditions in Beijing and its south area for a couple of days before reaching the sampling site. In this case, since haze is a regional phenomenon with similar meteorological conditions in Beijing and its south area (Zheng et al., 2015a; Zheng et al., 2015b; Wang et al., 2014), atmospheric parameters observed during haze at the sampling site should be typically representative for Beijing and its south area within the previous 2–3 days. As the reviewer mentioned, secondary sulfate formation in Beijing can be via the oxidation of atmospheric pollutants during transport and from local reactions. In the situation that secondary sulfate formation occurs during transport, since haze is a regional phenomenon with similar meteorological conditions in Beijing and its south area, our local atmospheric conditions-based calculations should be representative for secondary sulfate formation during transport, e.g., secondary sulfate formation within the previous 2-3 days in the south area. In fact, the overall sulfate production rate calculated on the basis of local atmospheric conditions increased from NPD to PD and basically coincided in time with the observed sulfate levels (Fig. 4 in the main text), which supports our local atmospheric conditions-based calculations being representative of secondary sulfate formation locally and during transport. To remind readers that our calculation is based on local atmospheric conditions rather than a regional reaction-transport model, we have changed the expression "our calculations" into "our local atmospheric conditions-based calculations" in the discussion and the abstract. We note that our local atmospheric conditions-based calculations may be not as robust as a regional reaction-transport model, which inspires future modelling work with the constraint of isotope data reported here to further improve the understanding of secondary sulfate

1

formation during Chinese haze. Therefore, we have added the expression "Our local atmospheric conditions-based calculations illustrate the utility of  $\Delta^{17}O(SO_4^{2^-})$  for quantifying sulfate formation pathways, but this estimate may be further improved with future regional modelling work." as a caveat in the abstract in lines 36-38.

**2. Q**:  $O_3$  is counted as one of the oxidants in the heterogeneous pathway in the manuscript (Eq. 14). This is debatable. It is very likely that in mineral dust surface or in aerosol water,  $O_3$  concentration can be negligible. If  $O_3$  is not counted, the estimated NO2 pathway would become much less important in the conclusion. In fact, this would be consistent with the possibility that NO2 may not be playing an important role in S(IV) oxidation after all, as some has suggested. At least, this point should be discussed in the manuscript.

A: Thanks for the comment. Our  $\Delta^{17}$ O observations are highly sensitive to ozone oxidation, and suggest a minor but significant role for this sulfate formation pathway. The  $\Delta^{17}$ O of sulfate produced via heterogeneous reactions ( $\Delta^{17}O_{hec}$ ) was calculated to be respectively 1.8 ‰, 3.1 ‰, 1.4 ‰, 0.1 ‰ and 0.8 ‰ for PD of Case I–V, which has been described in lines 308-309 in the present manuscript to replace the former expression "the  $\Delta^{17}$ O of sulfate produced via heterogeneous reactions ( $\Delta^{17}O_{hec}$ ) was calculated to range from 0.1 ‰ to 3.1 ‰ in our study." in the last manuscript. We also have added the description that "Since  $\Delta^{17}O(SO_4^{-2})$  produced via H2O2 oxidation is 0.7 ‰, smaller than  $\Delta^{17}O_{het}$  in Case I–III and V, O3 oxidation must contribute to heterogeneous sulfate production." in lines 309-311. As for whether O3 oxidation is negligible, our  $\Delta^{17}O$ -constrained calculation suggests heterogeneous O3 oxidation contributes 14 ‰, 11 ‰, 9 ‰, 11 ‰, 8 ‰, 0 % and 5 % under metastable assumption. The relatively high fraction in PD of Case I–II and low fraction in PD of Case III–V is consistent with the relatively high O3 values observed in PD of Case I–II and low O3 values observed in PD of Case III–V (Fig. 1c in the main text). Our  $\Delta^{17}O$ -constrained calculation should be more reliable than purely assuming O3 oxidation is negligible.

**3. Q:** Avoid claiming something like "the first observations of the oxygen-17 excess of..." in the manuscript. Many journals' instructions to authors specifically ask that you avoid using phrases like "we provide the first evidence" or "this is the first discovery". This is because it is more effective to

spell out how your work provides new knowledge and what important implications your discovery has. A: Thanks for the reminding. The word "first" has been removed from "the first observations of the oxygen-17 excess of..." and similar expressions throughout the manuscript.

4. Q: Mention the quantity (mg) of Ag2SO4 used for isotope measurement.

A: Thanks for the reminding. We have added the expression "The typical amount of  $O_2$  for each run is 0.4–0.8 µmol." in line 118 in the method, which corresponds to 125–250 µg of Ag2SO4.

**Reviewer 2 comments.**

**Q:** I appreciate the authors' detailed response to my comments. This is a very interesting paper utilizing D17O measurements to understand multiphase chemistry in Beijing haze. The clarity of this paper is improved. I only have a minor suggestion. As for the role of heterogeneous sulfate production (the main topic of this paper), Table S7 is very clear. I suggest moving this table to the main text. I agree that heterogeneous oxidation dominated in cases I, IV, and V. However, in case III, fcloud (46%) is slightly greater than fhet (42%). Therefore, both production pathways were important in this episode. I suggest the authors to present this in a careful way throughout the manuscript (including abstract). I recommend this paper to be published in ACP as it is, or with minor modification as suggested above. **A:** Thanks for the comment. Table S7 has been moved to the main text as Table 2. We have changed the former expression "However, heterogeneous sulfate production (*P*, ) on aerosols was estimated to

the former expression "However, heterogeneous sulfate production ( $P_{het}$ ) on aerosols was estimated to dominate sulfate formation during PD of other cases, with a fractional contribution of (48±5) %." in the abstract into "During PD of Case I and III–V, heterogeneous sulfate production ( $P_{het}$ ) was estimated to contribute 41–54 % to total sulfate formation with a mean of (48±5) %." in line 24-26. And we have changed the former expression "Heterogeneous reactions were found to dominate sulfate formation during PD in four out of the total five cases (except for Case II) with fractional contributions of 42 to 54 % and a mean of (48±5) % (Fig. 4)." in the discussion into "Heterogeneous reactions were found to contribute 41–54 % to total sulfate formation during PD of Case I and III–V, with a mean of (48±5) % (Fig. 4)." in lines 295-297.

**Editor Comments:**

**Dear authors,**

Many thanks for your revised submission. Please take note of the reviewers' comments and my

comments below when you prepare a revised manuscript.

**A:** Many thanks for your notice and comments. We prepare the revised manuscript following the reviewers' comments and your comments, and reply to these comments one by one.

**Q:** There are various problems with the dimensions of quantities in equations 1, 4, 5 and 8, and in case of equations 5 and 8, this may result in large changes to some of your results.

**A:** Thanks for your comment. We have corrected errors in these equations and errors in results from these equations. We reply to this comment in detail under your following specific comments.

**Q:** Section 2.7 does not state how the various fractions *f* have been calculated. Please give explicit equations that clarify this, including what the input terms (reaction rates,  $\Delta$ (17O) values) are and how these input terms have been calculated themselves.

A: Thanks for your comment. We have added how the various fractions f was calculated in the present manuscript. It reads "By using Eq. (6) and the definition  $f_{S(IV)+O3} + f_{S(IV)+H2O2} + f_{zero-d17O} = 1$ , we have  $f_{S(IV)+O3} = (d^{17}O_{obs}-0.7\%\times f_{S(IV)+H2O2})/6.5\%$  and  $f_{zero-d17O} = (6.5\%-d^{17}O_{obs}-5.8\%\times f_{S(IV)+H2O2})/6.5\%$ . Since  $f_{S(IV)+O3}$ ,  $f_{S(IV)+H2O2}$ , and  $f_{zero-d17O}$  should be in the range of 0 to 1 at the same time,  $f_{S(IV)+H2O2}$  is further limited to meet  $f_{S(IV)+H2O2} < \min\{d^{17}O_{obs}/0.7\%, (6.5\%-d^{17}O_{obs})/5.8\%\}$ . Therefore, possible range of  $f_{S(IV)+O3}$  and  $f_{zero-d17O}$  can be obtained at different  $f_{S(IV)+H2O2}$  assumptions." in lines 175-179 and " $f_p = c(p-SO_4^{2-})/c(SO_4^{2-})$ ,  $f_{het} = \{P_{het}/(P_{het}+P_{cloud}+P_{SO2+OH})\} \times (1-f_p)$ ,  $f_{cloud} =$  $\{P_{cloud}/(P_{het}+P_{cloud}+P_{SO2+OH})\} \times (1-f_p)$  and  $f_{SO2+OH} = \{P_{SO2+OH}/(P_{het}+P_{cloud}+P_{SO2+OH})\} \times (1-f_p)$ ." in lines 184-186, where  $c(p-SO_4^{2-})$  refers to the mass concentration of primary sulfate.

**Q:** The reviewers have also raised a number of points that need to be addressed. In particular, points 1 and 2 of reviewer 2 (source of the air; role of  $O_3$  as oxidant) need some careful discussion. Table S7 should be moved to the main text.

A: Thanks for your reminding. We have replied to the reviewers' comments in the above section.

Q: There are still a few problems with missing units in equations – please refer to the first chapter of the IUPAC Green Book (https://www.iupac.org/fileadmin/user\_upload/publications/e-resources/ONLINEIUPAC-GB3-2ndPrin

ting-Online-Sep2012.pdf) or chapter 5 of the SI brochure (https://www.bipm.org/en/publications/si-brochure/) for examples of correct quantity notation.

**A:** Thanks very much for recommending these books. We have corrected errors in the equations throughout the manuscript.

**Q:** The term "concentration" is not interchangeable with "mole fraction". Please use the term "mole fraction" where you refer to the latter (e.g. l. 197 and 198).

A: Thanks for your comment. Throughout the manuscript, we have changed "concentration" into "mole fraction" where it refers to the latter.

**Q:** Data availability: Please include a table with the data from Figures 1, 4, 5 and 6 and the individual input values used for each sample in the ISORROPIA model.

**A:** Thanks for your reminding. The data from Figures 1, 4, 5 and 6 and the individual values used for each sample in the ISORROPIA model are now available in the supplementary Excel file.

**Q:** 1. 67: A quantity symbol (e.g. *R*) must be used to define the isotope ratios and the index must follow immediately after the quantity symbol, e.g.  $R_{sample}(^{x}O/^{16}O)$ . "*X*" should be written in italics because it is a quantity symbol.

A: Thanks for your comment. We have changed the former expression "wherein  $\delta^{X}O = ((^{X}O/^{16}O)_{sample}/(^{X}O/^{16}O)_{VSMOW} - 1)$  with X = 17 or 18 and VSMOW referring to Vienna Standard Mean Ocean Water" into "wherein  $\delta = (R_{sample}/R_{reference} - 1)$  with *R* representing the isotope ratios of 17O/16O or 18O/16O in the sample and the reference Vienna Standard Mean Ocean Water, respectively" in lines 66-68 in the introduction.

**Q:** 1. 92: Please replace "ppb" with the corresponding SI unit "nmol mol-1", throughout the manuscript. Atmospheric Chemistry and Physics requires the use of SI units. Also, please write the equation in line with the rules of quantity algebra, i.e.  $[H_2O_2] / (nmol mol^{-1}) = 0.1155e^{0.0846T/C}$ .

**A:** Thanks for your suggestion. We have replaced "ppb" with "nmol mol-1" throughout the manuscript, and have rewritten the equation as you suggested in line 92.

Q: 1. 125/Eq. 1: Please use quantity algebra for all equations, see IUPAC Green Book

5

(https://www.iupac.org/fileadmin/user\_upload/publications/e-resources/ONLINEIUPAC-GB3-2ndPrin ting-Online-Sep2012.pdf). Where quantities are given as explicit values, they must carry units (e.g. 96 g mol-1, 3600 s h-1).

A: Thanks for your reminding. The quantities are with units now throughout the manuscript when they are given as explicit values.

**Q:** Eq. 1 is dimensionally not correct; it has units of g  $m^{-3} h^{-1} atm^{-1}$ , but is supposed to have g  $m^{-3} h^{-1}$ . Presumably the equation needs to include atmospheric pressure.

A: Thanks for your comment. Atmospheric pressure is included in equation 1 and 4 now. And we have corrected results from equation 1 and 4 throughout the manuscript. The following shows how we get equation 1 (similar for equation 4): The rate constant k ( $s^{-1}$ ) for heterogeneous loss of SO2 is determined by  $k = (R_p/D_g+4/v\gamma)^{-1}S_p$  (Jacob, 2000), therefore, the heterogeneous sulfate production rate  $P_{het} = k[SO_2]$  is in the unit of nmol (mol of air)-1 s-1 as the unit of SO2 (nmol mol-1) is indeed nmol (mol of air)-1. 1 nmol SO2 heterogeneous loss equals to 96 ng SO4-2 heterogeneous production, 1 s equals to 1/3600 h and 1 mol of air equals to the volume of  $(1mol \times RT)/p$  by using the ideal-gas law, so 1 nmol (mol of air)-1 s-1 = 96 ng  $((1mol \times RT)/p))^{-1}$  (1/3600 h)-1. When *R* is 0.082 atm L K-1 mol-1, *p* is in unit of atm and *T* is in the unit of K,1 nmol (mol of air)-1 s-1 = 96 ng  $(\frac{1mol \times RT}{p})^{-1}$  ( $\frac{1}{3600}$  h)-1 =  $\frac{3600 \times 96 \times (p)}{0.082(T)}$  µg m-3 h-1, where {*Q*} refers to the numerical value of a physical quantity *Q*. During our sampling period, the atmospheric pressure *p* ranged from 0.98 to 1.01 atm with a mean of (1.00±0.01) atm, so the corrected *P*het and *P*SO2+OH are both in the range of 98 % to 101 % of the former numerical values.

**Q:** 1. 127: The non-SI unit "atm" should be replaced with an SI-accepted unit, e.g. bar or Pa (or a derivative of them).

A: Thanks for your reminding. The non-SI unit "atm" have been replaced with an SI-accepted unit "Pa" in line 129.

Q: 1. 129: The uptake coefficient has the unit "1"; it is not "unitless".

A: Thanks for your reminding. The express "unitless" has been changed into the unit "1" in line 131.

**Q:** 1. 132: Again, units are missing from this equation. Also, the quantity that "PM2.5" refers to must be identified, e.g.  $\gamma(PM_{2.5})$  or  $\rho(PM_{2.5})$ , if it is a mass concentration. Both symbols are not ideal because they clash with the uptake coefficient and the bulk density. Perhaps the uptake coefficient should be given a different symbol than  $\gamma$ .

A: Thanks for your reminding. The former expression " $R_p = (0.254 \times PM_{2.5} + 10.259) \times 10^{-9}$ " has been changed into " $R_p/m = (0.254c(PM_{2.5})/(\mu g m^{-3}) + 10.259) \times 10^{-9}$ " in line 134, where  $c(PM_{2.5})$  refers to  $PM_{2.5}$  mass concentrations in the unit of  $\mu g m^{-3}$ .

**Q:** 1.136: The quantity that PM2.5 refers to must be identified, e.g.  $\gamma$ (PM2.5). The extraneous factor 10-6 and the multiplication symbols (×) should eliminated from the equation.

A: Thanks for your comment. The quantity " $PM_{2.5}$ " in equation 2 refers to  $PM_{2.5}$  mass concentration, which has been replaced by " $c(PM_{2.5})$ " now.

Q: l. 137 & 205: Please choose a suitable single-letter symbol for relative humidity in these equations,
e.g. Ψ.

A: Thanks for your comment. The expression "RH" has been changed into " $\Psi$ " in lines 140 and 212. And " $\Psi$ " has been identified as "where  $\Psi$  refers to relative humidity with the unit of %." in line 141.

**Q:** 1. 152: see 1. 125: "3600 s h-1", "96 g mol-1"; correct dimensions (presumably multiplication by atmospheric pressure).

A: Thanks for your comment. Atmospheric pressure is included in equation 4 now. " $3600 \text{ s h}^{-1}$ , 96 g mol-1," are used to replace "3600" and "96" in line 157.

**Q:** 1. 160 & 246: These equations is wrong in a bad way. The units on the right hand side are " $g^2 m^{-3} h^{-1}$ ", but are supposed to be " $g m^{-3} h^{-1}$ ". Again, it should be "3600 s h-1" and "96 g mol-1". Finally, the SI requires quantity symbols to consist of a single (Latin or Greek) letter, so LWC is not an acceptable symbol in an equation and should be replaced by a suitable one. (LWC as an abbreviation is fine, just not as a quantity symbol). These errors suggest that *P*cloud values may be fundamentally wrong. Please discuss, using numerical examples, the impact of correcting the equation on your results.

A: Thanks for your comment. Equation 5 is derived from subsection 7.4 of Seinfeld and Pandis (2006)

at pp. 306, which reads:

The moles per liter of air can be then converted to equivalent  $SO_2$  partial pressure for 1 atmosphere total pressure by applying the ideal-gas law to obtain

$$R_a'' = 3.6 \times 10^6 \, LRT \, R_a \quad (\text{ppb h}^{-1}) \tag{7.75}$$

where L is in g m-3, R = 0.082 atm L K-1 mol-1, and T is in K. For example, an aqueousphase reaction rate of 1  $\mu$ M s-1 in a cloud with a liquid water content of 0.1 g m-3 at 288 K is equivalent to a gas-phase oxidation rate of 8.5 ppb h-1. A nomogram relating aqueous-

It needs to be explained that  $R_a$  is the reaction rate in M s-1, L is cloud liquid water content in g m-3 in the above equation 7.75. Since 1 ppb  $h^{-1} = 1$  nmol (mol of air)-1  $h^{-1} = 96$  ng  $(\frac{1 \text{mol} \times RT}{p})^{-1} h^{-1} = 1$ 96 ng  $\left(\frac{1 \mod \times 0.082 \operatorname{atm} L \operatorname{K}^{-1} \operatorname{mol}^{-1} \times \{T\} \operatorname{K}}{\{p\} \operatorname{atm}}\right)^{-1} \operatorname{h}^{-1} = \frac{96 \times \{p\}}{0.082 \{T\}} \mu g \operatorname{m}^{-3} \operatorname{h}^{-1}$ , 8.5 ppb h-1 (an approximate value of 8.502 ppb  $h^{-1}$ ) in the above case equals to 34.56  $\mu$ g m-3  $h^{-1}$  of sulfate production under 1 atmosphere total pressure. When we calculate the case (LWC = 0.1 g m-3 = 100 mg m-3,  $R_{S(IV)+oxi} = 1 \mu M s^{-1} = 10^{-6}$ M s-1) by using our equation 5, it turns out  $P_{\text{cloud}} = 3600 \times 96 \times 100 \times 10^{-6} = 34.56$ , which is the same as value calculated by above equation 7.75. In fact, the unit of right hand side of our equation 5 is truly " $\mu$ g m-3 h-1" as 1 mg m-3 of LWC equals to 10-3 ml m-3 when  $\rho$ (H2O) = 1g ml-1 was used, which is also used by Seinfeld and Pandis (2006) to get the unit of ppb  $h^{-1}$  in the above equation 7.75. Here shows how we get the unit of  $P_{\text{cloud}}$  in equation 5 (similar for  $P_{\text{het}}$  in equation 18): (s h-1)×(g mol-1)×(mg m-1) 3)×(M s-1) = (s h-1)×(g mol-1)×(10-6 L m-3)×(mol L-1 s-1) = 10-6 g m-3 h-1 =  $\mu$ g m-3 h-1. For your review, we shows how to obtain equation 5:  $P_{\text{cloud}} = LWC \times R_{S(IV)+\text{oxi}} = \{LWC\}(\text{mg m}^{-3}) \times \{R_{S(IV)+\text{oxi}}\}(M \text{ s}^{-1}) = \{LWC\}(M \text{ s}^{-1})$  $\{LWC\} \times \{R_{S(IV)+oxi}\} (mg m^{-3}) \times (M s^{-1}) = \{LWC\} \times \{R_{S(IV)+oxi}\} (10^{-6} L m^{-3}) \times (mol L^{-1} s^{-1}) = 0$ refers to the numerical value of Q. In addition, we have changed "LWC" and "AWC" into " $L_c$ " and " $L_a$ " in equations 5 and 18 respectively, to meet the requirements of SI.

**Q:** 1. 167 & 1. 180: Include "%" after 6.5 and 0.7.**

**A:** Thanks for your reminding. We have added "‰" after 6.5 and 0.7 in equations 6 and 8 in lines 172 and 189.

**Q:** 1. 196: This equation requires quantity symbols for the mole fractions and the unit "nmol  $mol^{-1}$ "

needs to appear in the right place, e.g. " $y(NH_3) = 0.34y(NOx) + 0.63 \text{ nmol mol}^{-1}$ ".

A: Thanks for your suggestion. We have changed "NH3 (ppb) =  $0.34 \times NO_X$  (ppb) + 0.63" into "[NH3]/(nmol mol-1) =  $0.34[NO_X]/(nmol mol^{-1}) + 0.63$ " in line 203.

**Q:** 1. 197: Replace "concentration" with "mole fraction" – also other occurrences of the word "concentration" in the text may need to be replaced with "mole fraction". Concentration implies an amount per volume.

A: Thanks for your suggestion. Throughout the manuscript, we have changed "concentration" into "mole fraction" where it refers to the latter.

**Q:** 1. 205: "MF" should be replaced with a suitable single-letter symbol, e.g. "*x*(metastable)".

A: Thanks for your suggestion. "MF" has been replaced by "x(metastable)" in equation 9 in line 212 and identified as "where x(metastable) is the fraction of metastable aerosols to total aerosols in the unit of %." in line 213.

**Q:** 1. 211: The terms involving logarithms of concentrations and ion strengths in equations need to be divided by the standard concentration ( $c^{\ominus} = 1 \mod \text{dm}^{-3}$ ), to make them dimensionally correct. The units of  $\beta^*$  need to be identified.

A: Thanks for your comment. The terms involving logarithms of concentrations and ion strengths in equations have been divided by " $c \ominus$ " in line 218-222. The unit of  $\beta^*$  is (mol L-1)2, which has been identified in line 225.

**Q:** 1. 258: Please use appropriate symbols, e. g  $c(SO_4^{2-})$  for sulfate concentrations.

A: Thanks for your suggestion. c(X) has been used as the mass concentration of species X throughout the manuscript. The expression "SOR =  $nSO_4^{2-}/(nSO_4^{2-}+nSO_2)$ , where  $nSO_4^{2-}$  and  $nSO_2$  represents the molar concentration of  $SO_4^{2-}$  and  $SO_2$ , respectively" has been changed into "SOR, which equals to  $SO_4^{2-}$  molar concentration divided by the sum of  $SO_4^{2-}$  and  $SO_2$  molar concentration" in line 264-265.

**Q:** p. 11: Remove unnecessary brackets around pH expressions, e. g.  $7.6\pm0.1$ . The brackets are only required where similar such expressions have units.

A: Thanks for your reminding. Unnecessary brackets around pH expressions have been removed throughout the manuscript.

**Q:** Figure 2: In the figure caption, please include an explicit link to the newly added equations in the main text that give explicit solutions for the fractions shown in this figure.

A: Thanks for your suggestion. We have added the expression " $f_{S(IV)+H2O2}$  is in the range of 0 to  $min\{\Delta^{17}O_{obs}/0.7\%, (6.5\%-\Delta^{17}O_{obs})/5.8\%\}, f_{S(IV)+O3} = (\Delta^{17}O_{obs}-0.7\%\times f_{S(IV)+H2O2})/6.5\%$  and  $f_{zero-\Delta^{17}O} = (6.5\%-\Delta^{17}O_{obs}-5.8\%\times f_{S(IV)+H2O2})/6.5\%$ . See equation 6 and its caption in Sect. 2.7 for details." in the end of Figure 2 caption in lines 578-580.

**Q:** Table S1:  $k_{0\text{low}} = 3.3 \times 10^{-31} (T/300 \text{ K})^{-4.3} \text{ cm}^6 \text{ s}^{-1} [T \text{ has units of K; molecule is not a unit}]$  $k_{0\text{high}} = 1.6 \times 10^{-12} \text{ cm}^3 \text{ s}^{-1} [\text{since } (T/300 \text{ K})^0 = 1]$

A: Thanks for your suggestion. We have improved the expressions in Table S1 based on your suggestion.

**Q:** Table S4: Did you only use these mean values in your thermodynamic calculations? Or did you use sample-specific input parameters?

A: We used sample-specific input parameters. And these input has been presented in the supplementary Excel file.

**References**

Jacob, D. J.: Heterogeneous chemistry and tropospheric ozone, Atmos. Environ., 34, 2131-2159, 2000.

- Seinfeld, J. H., and Pandis, S. N.: Atmospheric chemistry and physics: From air pollution to climate change, John Wiley & Sons, New Jersey, 2006.
- Wang, Y., Zhang, Q., Jiang, J., Zhou, W., Wang, B., He, K., Duan, F., Zhang, Q., Philip, S., and Xie, Y.: Enhanced sulfate formation during China's severe winter haze episode in January 2013 missing from current models, J. Geophys. Res., 119, 10425-10440, 2014.
- Zheng, B., Zhang, Q., Zhang, Y., He, K., Wang, K., Zheng, G., Duan, F., Ma, Y., and Kimoto, T.: Heterogeneous chemistry: a mechanism missing in current models to explain secondary inorganic

aerosol formation during the January 2013 haze episode in North China, Atmos. Chem. Phys., 15, 2031-2049, 2015a.

Zheng, G., Duan, F., Su, H., Ma, Y., Cheng, Y., Zheng, B., Zhang, Q., Huang, T., Kimoto, T., and Chang, D.: Exploring the severe winter haze in Beijing: the impact of synoptic weather, regional transport and heterogeneous reactions, Atmos. Chem. Phys., 15, 2969-2983, 2015b.

---

## Author Response (AR3)

**Comments to the Author:**

Dear Dr Xie

Thank you for submitting your revision to Atmospheric Chemistry and Physics, and for your careful attention to the earlier reviews of this article and my own comments.

I think that your manuscript is nearly ready for acceptance, except for a small error in Equations 5 and 18, which are currently still dimensionally wrong. Based on your own explanations, the right hand side needs to be divided by the density of water ($\rho\_w$) and this needs to be shown by the equation. With $\rho\_w$ = 1000 kg m-3 and the suggested units as per your manuscript, the result should then be in µg m-3 h-1, so I think numerically this does not lead to any changes.

Incidentally, the equation of Seinfeld and Pandis (2006) is also wrong. It needs to be divided by the density of water and atmospheric pressure, to be dimensionally correct. I haven't got the the latest (third) edition of this textbook to hand, but if the equation is also wrong in the latest edition, I'd be happy to write to Seinfeld and Pandis and explain this.

Kind regards
Jan Kaiser
Editor Atmospheric Chemistry and Physics

**Responses to the Editor:**

Dear Dr Kaiser

Thanks very much for your careful comments. The right hand side of equation 5 and 18 is divided by the density of water ($\rho_w$) now and $\rho_w$ is identified as "$\rho_w$ is the density of water ($1\mathrm{kg}\ \mathrm{L}^{-1}$)." in line 167 and 255. As for the equation of Seinfeld and Pandis (2006), I think you are right. I'm sorry that I haven't got the third edition of this textbook either, so I'm not sure whether they have corrected this mistake.

Yours sincerely
Zhouqing Xie
On behalf of all co-authors